# Molecular basis promoting centriole triplet microtubule assembly

Yutaka Takeda [1], Takumi Chinen [1] ✉, Shunnosuke Honda[1], Sho Takatori[2], Shotaro Okuda[1], Shohei Yamamoto[1], Masamitsu Fukuyama[1], Koh Takeuchi [3], Taisuke Tomita [2], Shoji Hata [1] ✉ & Daiju Kitagawa [1] ✉

The triplet microtubule, a core structure of centrioles crucial for the organization of centrosomes, cilia, and flagella, consists of unclosed incomplete microtubules. The mechanisms of its assembly represent a fundamental open question in biology. Here, we discover that the ciliopathy protein HYLS1 and the β-tubulin isotype TUBB promote centriole triplet microtubule assembly. HYLS1 or a C-terminal tail truncated version of TUBB generates tubulin-based superstructures composed of centriole-like incomplete microtubule chains when overexpressed in human cells. AlphaFold-based structural models and mutagenesis analyses further suggest that the ciliopathy-related residue D211 of HYLS1 physically traps the wobbling C-terminal tail of TUBB, thereby suppressing its inhibitory role in the initiation of the incomplete microtubule assembly. Overall, our findings provide molecular insights into the biogenesis of atypical microtubule architectures conserved for over a billion years.

The centriole, an ancient organelle conserved for over a billion years, is essential for the formation of cilia, flagella, and centrosomes[1–4]. It is characterized by triplet microtubules arranged in a nine-fold symmetry. Each triplet microtubule is composed of one complete tubule (A-tubule) and two additional incomplete tubules (B- and C-tubules)[4–7]. The unclosed B- and C-tubules nucleate from the outer junctions on the lateral surfaces of the A- and B-tubules, respectively[5–7]. This atypical microtubule structure with incomplete tubules is widely considered to be linked with the unique properties and functions of the centrioles, but the mechanisms of its assembly remain a mystery.

The triplet microtubules are crucial for the centrioles to organize cilia, flagella, and centrosomes[8,9]. The centriolar A- and B-tubules serve as templates for the doublet microtubules of cilia and flagella, thereby providing structural integrity and basis for dynein-dependent transport[1,2]. In human cells, centrioles lacking the incomplete tubules fail to accumulate centrosomal proteins required for microtubule nucleation and therefore cannot assemble functional centrosomes[8]. Furthermore, such aberrant structures do not display high stability, which is a hallmark property of the centrioles[8,10]. Although several

centriolar proteins are reported to contribute to the structural integrity of incomplete tubules[8–10], key factors that directly promote their assembly have not been identified.

In the 1980s, in vitro experiments using the bacterial protease subtilisin successfully reconstituted incomplete microtubules that resemble those observed within centrioles[11]. Subtilisin enables purified tubulin dimers to assemble chains of incomplete microtubules by cleaving their C-terminal tails (CTTs)[11]. A more recent study further achieved the in vitro reconstitution of doublet microtubules, where incomplete B-tubules were nucleated on the lateral surfaces of subtilisin-treated complete A-tubules[12]. Molecular dynamic simulations show that the intrinsically disordered CTTs of the A-tubule generate spatial interferences for the attachment of B-tubule protofilaments[12], indicating that the artificial removal of CTTs by subtilisin can mimic the nucleation of incomplete tubules from the lateral surfaces of microtubules in cells. However, in vivo regulators of the tubulin CTTs in the process of centriole triplet microtubule assembly have not been determined.

[1]Laboratory of Physiological Chemistry, Graduate School of Pharmaceutical Sciences, The University of Tokyo, Bunkyo, Tokyo 113-0033, Japan. [2]Laboratory of Neuropathology and Neuroscience, Graduate School of Pharmaceutical Sciences, The University of Tokyo, Bunkyo, Tokyo 113-0033, Japan. [3]Laboratory of Physical Chemistry, Graduate School of Pharmaceutical Sciences, The University of Tokyo, Bunkyo, Tokyo 113-0033, Japan. ✉e-mail: takumi.chinen@mol.f.u-tokyo.ac.jp; s.hata@mol.f.u-tokyo.ac.jp; dkitagawa@mol.f.u-tokyo.ac.jp

In this study, we identify the ciliopathy protein HYLS1 as a key regulator, which modulates the tubulin CTT to promote the assembly of centriole triplet microtubules in human cells. Overexpression of HYLS1 generates tubulin-based superstructures with incomplete microtubules that require the β-tubulin isotype TUBB. Furthermore, AlphaFold-based mutagenesis analyses suggest that the ciliopathy-related residue D211 of HYLS1 interacts with the CTT of TUBB and suppresses its inhibitory role in incomplete tubule formation. Overall, our study discovers a core mechanism of centriole triplet microtubule assembly, providing a critical insight into the long-standing question in biology.

## Results

### Overexpression of the ciliopathy protein HYLS1 leads to assembly of centriole-like stable superstructures with incomplete microtubule chains

To identify proteins that provide the stability of centrioles, we screened for genes with similar properties to those already identified as centriole stabilizing factors (*CEP295* and *CEP44*)[8]. Since centrioles play crucial roles in cell division and survival[13], we utilized indexes of essentiality for cell proliferation as a measure of gene properties. The Cancer Dependency Map (DepMap) catalogues almost all human genes tied to their essentiality level (Chronos scores) assessed in CRISPR-Cas9-based screens across over 1000 human cell lines[14,15]. In this database, the Chronos scores of *CEP295* and *CEP44* vary widely across different cell lines in contrast to those of centriole duplication factors (*PLK4*, *STIL*, and *SASS6*)[16–19] that are universally essential for cell proliferation (Fig. 1a). The Chronos scores of *CEP295* and *CEP44* showed strong correlation (Fig. 1b; $r = 0.280$), suggesting that other genes required for the stability of centrioles may follow a similar trend.

To identify genes that are highly correlated with *CEP295* and *CEP44* in Chronos scores, we conducted a two-step screen (Fig. 1c). In the first step, we extracted the top 100 hits from 17393 human genes according to the averages of correlation coefficients of their Chronos scores with those of *CEP295* and those of *CEP44* (Fig. 1d). In the second step, we performed a hierarchical clustering analysis of the top 100 genes based on their Chronos scores (Fig. 1e). This analysis categorized *CEP295*, *CEP44*, and other genes involved in centriole formation (*CEP120* and *C2CD3*)[20,21] into the same small cluster. This cluster also included the ciliopathy-related gene *HYLS1* which encodes a centriolar protein necessary for the formation of intact cilia in *C. elegans*, *D. melanogaster*, *Xenopus* embryos, and human cells[22–26]. Our stimulated emission depletion (STED) microscopy imaging of HeLa mCherry-HYLS1 knock-in cells (Supplementary Fig. 1a–f) confirmed that endogenous HYLS1 is localized to the centrioles (Fig. 1f). However, the exact role of HYLS1 in centriole stabilization has not been characterized in human cells.

To gain insights into the function of HYLS1 in centriole stabilization, we overexpressed HYLS1 in RPE-1 cells and observed that overexpressed HYLS1 formed large rod-like structures, preserved by cross-linking paraformaldehyde (PFA) fixation (Fig. 1g; PFA fixation). These superstructures were positive for anti-α-tubulin immunostaining, indicating that they are tubulin-based. After methanol (MeOH) fixation, the tubulin-based superstructures could be observed more clearly, but they no longer appeared co-localizing with the HYLS1 signal that displayed a distinct, network-like organization (Fig. 1g; MeOH fixation). Fluorescence live-cell imaging upon mScarlet-i-HYLS1 overexpression confirmed that HYLS1 and tubulin co-assemble in the same superstructures (Fig. 1h, Supplementary Fig. 2a, Supplementary Movie 1). Thus, MeOH fixation likely dissociates HYLS1 from the tubulin-based superstructures. A closer look at the live-cell imaging data revealed that the tubulin-based superstructure followed the appearing and elongation of the HYLS1 rod-like structure with a certain delay (Fig. 1h). Similar tubulin-based superstructures were observed upon HYLS1 overexpression in another human cell line (Supplementary Fig. 2b). HYLS1 homologs of *C. elegans* and *D. melanogaster* were

also able to generate similar superstructures when overexpressed in RPE-1 cells (Supplementary Fig. 2c). Therefore, HYLS1 promotes the formation of these tubulin-based superstructures through an evolutionarily conserved function.

Because the tubulin-based superstructures resembled centrioles, we tested whether they have similar properties. Centrioles are highly stable cylindrical assemblies composed of microtubules that harbor characteristic tubulin posttranslational modifications and recruit various specific proteins[3,27–29]. STED imaging revealed that the HYLS1-induced tubulin-based superstructures also exhibit cylindrical shapes, but their average diameter (486 nm) was larger than that of the centrioles (205 nm) (Fig. 1i, j). No centriole-associated proteins or tubulin posttranslational modifications were observed at the tubulin-based superstructures, except for tubulin acetylation, which is a marker for stable microtubules (Supplementary Fig. 3a–c). Notably, the superstructures exhibited high stability similar to centrioles in assays with low temperature and nocodazole treatment which normally disassemble cytoplasmic microtubules[27,28] (Fig. 1k–m and Supplementary Fig. 3d–f). Taken together, the HYLS1-induced tubulin-based superstructures are different from centrioles in several aspects, but similar in terms of stability and resistance to depolymerization.

To reveal the structural features of the tubulin-based superstructures, we employed correlative light and electron microscopy (CLEM) and successfully captured superstructures in HYLS1 overexpressed cells (Supplementary Fig. 4a). In the wall of the vertically oriented (Top-view) superstructure, a particular motif was observed with a string of linked unclosed tubules, each protruding from the lateral surface of the adjacent tubule at approximately equal intervals (Fig. 2a and Supplementary Fig. 4b). In addition, the motif was also observed in a seed-like structure, which, unlike the cylindrical structures, did not form a circumference (Supplementary Fig. 4c). This motif resembles the incomplete tubule chains observed in doublet and triplet microtubules reconstituted in vitro[11,12]. The horizontal superstructures (Side-view) displayed inner and outer layers on both sides of the lateral surfaces (Fig. 2b and Supplementary Fig. 4b). The average width between the two layers (interlayer distance; 14.4 nm) was significantly smaller than that of singlet microtubules (20.0 nm) (Fig. 2c), but consistent with the previously reported average width of native ciliary incomplete microtubules (14.6 nm)[30]. Based on these structural features, we hereafter refer to the superstructure as incomplete MT superstructure. These results indicate that HYLS1 generates incomplete microtubule chains similar to those of centriole triplet microtubules (Fig. 2d).

### HYLS1 provides the structural integrity of centriole triplet microtubules

To assess the role of HYLS1 for the structural integrity of centriole triplet microtubules, we depleted the protein in RPE-1 cells using RNAi (Supplementary Fig. 5a). First, using TEM, we observed the ultrastructure of younger centrioles that formed in the preceding cell cycle (Supplementary Fig. 5b). While centrioles in control cells showed nine intact triplet microtubules, HYLS1-depleted cells exhibited severely or partially broken centriole structures (Fig. 2d). These defective centrioles are similar to those lacking the B- and C-tubules shown in previous studies[10,31], suggesting that HYLS1 is necessary for the assembly of these incomplete microtubules within the centrioles. In support of this, in previous studies using *C. elegans*, HYLS1 was revealed to be crucial for the structural integrity of cilia with doublet microtubules[23,24]. In contrast, centrioles in *C. elegans* early embryos, which are composed of singlet microtubules, do not require HYLS1 for their assembly[23].

Centrioles lacking incomplete microtubules had displayed significant defects in elongation and stabilization in previous studies using human cells[8,10,31]. Consistent with this, we found that the average length of procentrioles in HYLS1-depleted cells (143 nm) was significantly shorter than that in control cells (365 nm) (Supplementary

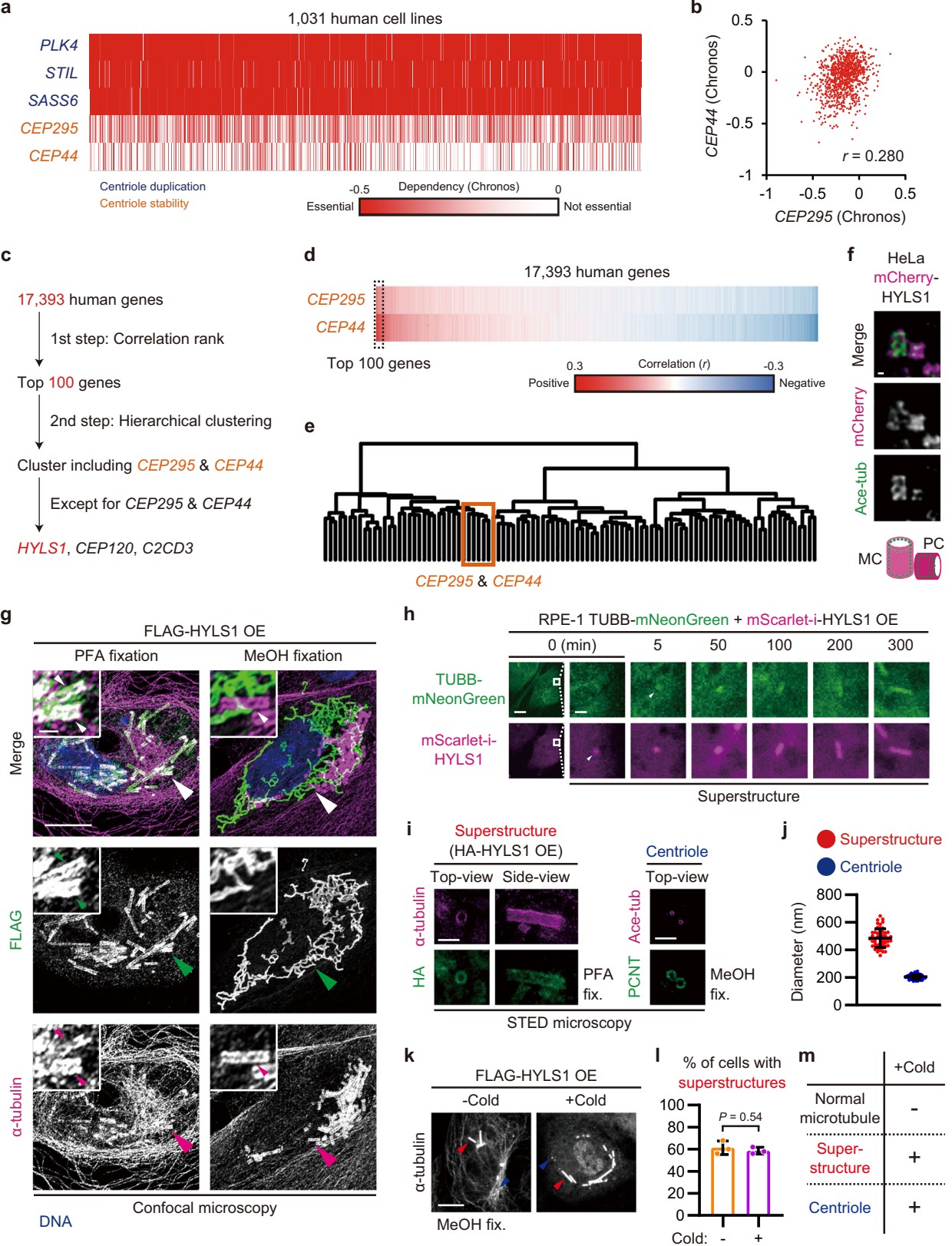

Fig. 5c, d). We also evaluated the stability of centrioles using STLC[32] and cold treatment[8] (Supplementary Fig. 5e, f). In both assays, the centrioles in HYLS1-depleted cells were less stable than those in control cells (Fig. 2e, f and Supplementary Fig. 5g–j). Taken together, HYLS1 provides the structural integrity of centrioles with intact triplet microtubules in human cells.

## HYLS1 ensures the ability of centrioles to organize centrosomes and primary cilia

The structural integrity of centriole triplet microtubules is important for the formation of centrosomes and primary cilia[8,9]. Centrioles are converted into centrosomes through the stepwise recruitment of centrosomal proteins that enable them to nucleate microtubules and

**Fig. 1 | Overexpression of the ciliopathy protein HYLS1 leads to assembly of tubulin-based superstructures with high stability similar to that of centrioles.**
**a** Heatmap of Chronos scores of the indicated genes for 1031 human cell lines.
**b** Scatter plots of Chronos scores of *CEP295* and *CEP44*. **c** Schematic of the two-step screening. **d** Heatmap of correlation coefficients between Chronos scores of the indicated genes and those of 17,393 human genes. The genes were sorted in order of mean values. **e** Hierarchical clustering analysis of the top 100 hits using their Chronos scores. The cluster enclosed by orange rectangle included *CEP295*, *CEP44*, *CEP120*, *C2CD3*, and *HYLS1*. **f** STED microscopy images of HeLa mCherry-HYLS1 knock-in cells. Scale bar: 100 nm. MC: mother centriole, PC: procentriole. Ace-tub: acetylated tubulin. **g** Confocal microscopy images of RPE-1 cells transfected with pCMV-FLAG-HYLS1 and fixed with the indicated methods. Scale bars: 10 μm and 1 μm. Large arrowheads: magnified areas, small arrowheads: superstructures.

**h** Time-lapse images of RPE-1 cells stably expressing TUBB-mNeonGreen transfected with pCMV-mScarlet-i-HYLS1. Scale bars: 10 μm and 1 μm. Arrowheads: first confirmed accumulations. **i** STED microscopy images of the tubulin-based superstructures in RPE-1 cells transfected with pCMV-HA-HYLS1 and centrioles in RPE-1 cells. Scale bars: 1 μm. Ace-tub: acetylated tubulin. **j** Quantification of diameter of the tubulin-based superstructures and the centrioles in (**i**). $n = 50$ structures. **k** IF images of RPE-1 cells transfected with pCMV-FLAG-HYLS1 and then incubated on ice for 1 h before fixation. Scale bar: 10 μm. Red arrowheads: superstructures, blue arrowheads: centrioles. **l** Quantification of frequency of interphase cells with the tubulin-based superstructures in (**k**). $n = 3$ independent experiments, 50 cells each. **m** Schematic showing whether each structure remains at low temperature. Data are represented as mean ± s.d. *P* value was calculated by two-tailed unpaired Student's *t*-test (**l**). Source data are provided as a Source Data file.

to duplicate themselves[33–35]. For newly formed (younger) centrioles, this process occurs from late mitosis to the next G1 phase to maintain the numbers of both centrioles and centrosomes in the subsequent cell cycle. In HYLS1-depleted cells, younger centrioles failed to recruit key centrosomal proteins[36] (Supplementary Fig. 6a–h) and displayed significantly reduced microtubule nucleation activity (Fig. 2g, h). Moreover, long-term depletion of HYLS1 reduced the numbers of centrioles and centrosomes (Supplementary Fig. 6i–k). The microtubule nucleation activity and number of centrosomes are both crucial for proper mitotic spindle formation[13]. While most of the control cells established bipolar mitotic spindles with two centrosomes, a significant proportion of HYLS1-depleted cells showed abnormal mitotic spindles (Fig. 2i, j). Centrioles also serve as a basis for the assembly of primary cilia for signal transduction[1,2]. We observed a significant decrease in ciliated cells and shortening of primary cilia upon HYLS1 depletion (Fig. 2k–m). These data indicate that HYLS1 provides the structural integrity of centriole triplet microtubules to organize centrosomes and primary cilia (Fig. 2n).

### The ciliopathy-related D211G mutant of HYLS1 fails to assemble incomplete microtubules and stable centrioles

To further understand the molecular properties of HYLS1, we mapped the domains required for the assembly of incomplete MT superstructures. To this end, we overexpressed several truncated forms of HYLS1 and found that the C-terminal half (151-299 a.a.) of HYLS1 is sufficient for the assembly of incomplete MT superstructures (Fig. 3a–c). The C-terminal half of HYLS1 contains an evolutionarily conserved domain (HYLS-1 Box), where the aspartic acid at position 211 (D211) is often mutated to glycine (G) in patients with hydrolethalus syndrome, a form of severe ciliopathy[22] (Fig. 3d). Although the D211G mutant of HYLS1 itself formed structures similar to those of the wild-type, it completely failed to assemble incomplete MT superstructures (Fig. 3e, f). To verify whether the ability of HYLS1 to assemble incomplete microtubules is required for the structural integrity of centrioles, we performed rescue experiments with the D211G mutant in HYLS1-depleted cells. Using cell lines which express RNAi-resistant wildtype or D211G mutant of HYLS1 under the inducible Tet-On system (Supplementary Fig. 7a), we analyzed the centriole stability upon depletion of endogenous HYLS1 and doxycycline treatment. In both stability assays with STLC and cold treatment, the D211G mutant of HYLS1 could not rescue the centriole destabilization, while the centriole structures persisted when the wildtype was expressed (Fig. 3g, h and Supplementary Fig. 7b–d). These data suggest that HYLS1, through its ciliopathy-related residue D211, assembles incomplete microtubules to ensure the structural integrity of centrioles.

### The β-tubulin isotype TUBB is specifically required for the HYLS1-dependent assembly of incomplete microtubules

Given that HYLS1 assembles incomplete MT superstructures when overexpressed, it may directly interact with tubulin. To test this

possibility, we performed in silico analysis of physical interaction potentials between HYLS1 and tubulin. Since human has eight α- and nine β-tubulin isotypes[37], we generated structural models of the protein complexes of HYLS1 with each tubulin isotype using AlphaFold-Multimer[38,39] (Fig. 4a). AlphaFold-Multimer provided structural predictions in which HYLS1 wraps around specific tubulin isotypes (TUBA1A and TUBB/TUBB5) (Fig. 4b, c). The predicted aligned error (PAE), a metric for the confidence in the relative positions of the predicted structures, shows low values in the models of HYLS1 and the two tubulin isotypes, thereby indicating the plausibility of their direct interactions (Fig. 4b, c). Through immunoprecipitation, we experimentally confirmed that HYLS1 physically interacts with TUBB (Fig. 4d). Together the calculations and the experimental validation show that HYLS1 can directly interact with tubulin.

Next, to identify the tubulin isotypes required for the HYLS1-dependent assembly of incomplete microtubules, we systematically knocked down each tubulin isotype and then overexpressed HYLS1 in RPE-1 cells. Depletion of any α-tubulin did not affect the assembly of incomplete MT superstructures (Fig. 4e, f and Supplementary Fig. 8a–c). On the other hand, depletion of TUBB specifically suppressed the formation of these superstructures (Fig. 4e, f and Supplementary Fig. 8a, b, d). The localization of TUBB to incomplete MT superstructures was confirmed via the live-cell imaging (Fig. 1h and Supplementary Fig. 2a). These data indicate that HYLS1 interacts with TUBB to assemble incomplete MT superstructures.

### TUBB provides the structural integrity of centriole triplet microtubules

To investigate whether TUBB is also specifically required for the structural integrity of centrioles, we analyzed their stability upon depletion of each tubulin isotype. In the stability assay with STLC treatment, TUBB depletion specifically destabilized centrioles among all the tubulin isotypes (Fig. 5a, b and Supplementary Fig. 9a–c). Whereas younger centrioles in control cells showed nine intact triplet microtubules, about half of the younger centrioles in TUBB-depleted cells displayed broken structures (Fig. 5c and Supplementary Fig. 10a). Similar to the observation in HYLS1-depleted cells, the recruitment of key centrosomal proteins to younger centrioles was partially disrupted upon TUBB depletion (Supplementary Fig. 10b–i). In addition, a significant decrease in ciliated cells and shortening of primary cilia were equally detected in TUBB-depleted cells (Fig. 5d–f). In support of the importance of TUBB at centrioles, TUBB-FLAG expressed by the Tet-On system localized to centrioles (Supplementary Fig. 10j). These data indicate that TUBB provides the structural integrity of centriole triplet microtubules and ensures the formation of centrosomes and primary cilia in human cells.

### The unstructured C-terminal tail of TUBB prevents the assembly of incomplete microtubules

To understand the molecular mechanism through which HYLS1 assembles incomplete microtubules in a TUBB-dependent manner, we

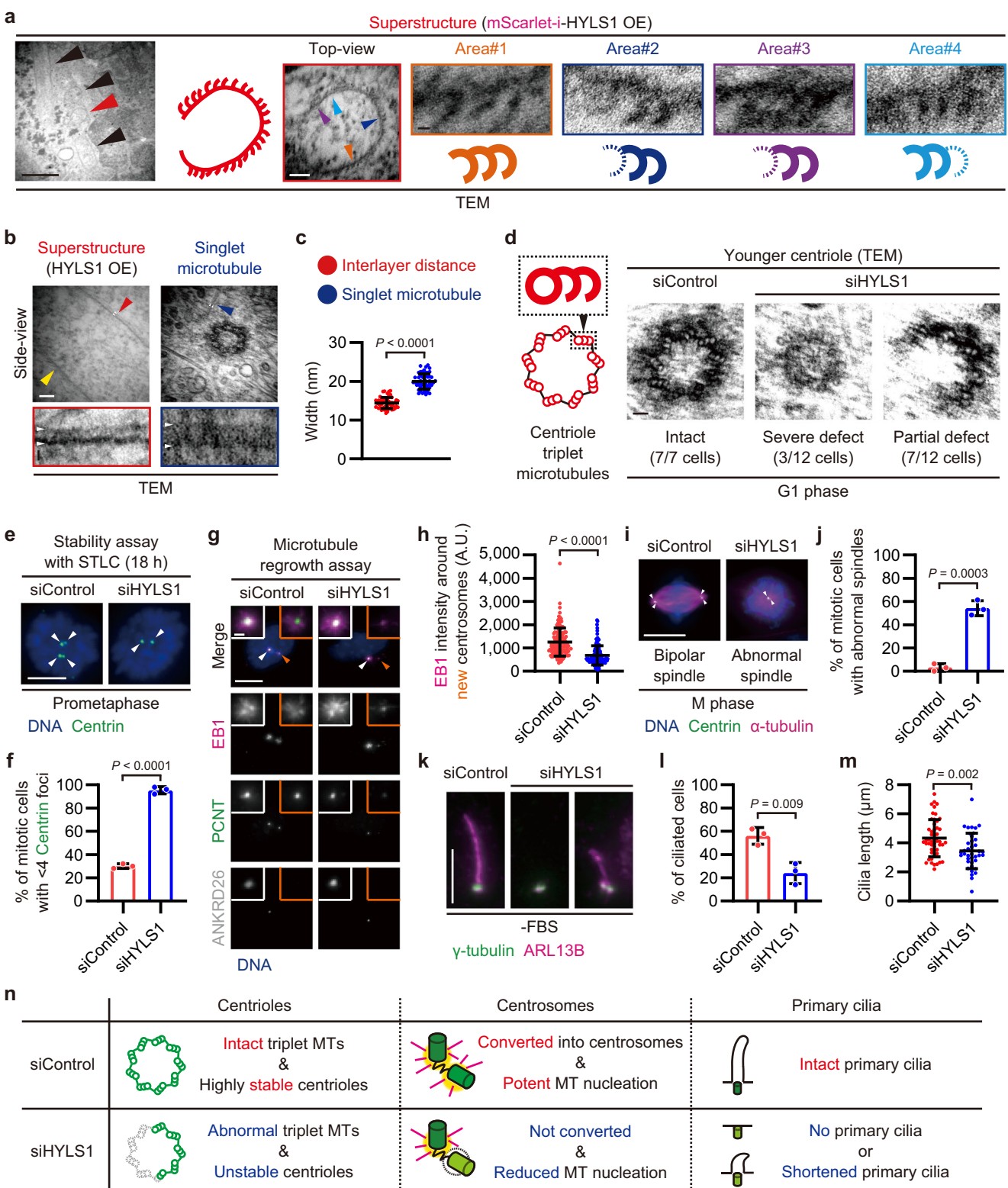

focused on the structural models of the HYLS1-TUBB complex generated by AlphaFold-Multimer (Fig. 4c). In the models, the HYLS1 D211 residue required for the assembly of incomplete microtubules was in close proximity to the C-terminal tail (CTT) of TUBB (Fig. 6a), predicting their physical interaction. The tubulin CTTs, which show a significant variation in their amino acid compositions among the otherwise highly conserved tubulin isotypes (Fig. 6b), have been shown to play an inhibitory role for the assembly of incomplete

microtubules in vitro[11,12]. Previous in silico analyses indicate that the intrinsically disordered CTTs of existing microtubules create spatial interferences for the attachment of incomplete tubule protofilaments to the lateral surfaces of the microtubules[12]. The presence of HYLS1 in our models decreased the flexibility of the TUBB CTT among the five models (Fig. 6c), indicating that their interaction could reduce the spatial interferences to allow incomplete microtubules to nucleate from the lateral surfaces of existing microtubules.

**Fig. 2 | HYLS1 provides the structural integrity of centriole triplet microtubules to organize centrosomes and primary cilia. a** TEM images of vertically oriented superstructures in RPE-1 cells transfected with pCMV-mScarlet-i-HYLS1. Scale bars: 1 µm, 100 nm, and 10 nm. Large arrowheads (red & black): superstructures, small arrowheads (orange, blue, purple, and cyan): magnified areas in the wall of the superstructure highlighted in red. **b** TEM images of horizontal superstructures in RPE-1 cells transfected with pCMV-mScarlet-i-HYLS1 and singlet microtubules in RPE-1 cells. Scale bars: 100 nm and 10 nm. Large arrowheads: both sides of the lateral surface of the horizontal superstructure (red and yellow) or magnified areas (red and blue), small arrowheads (white): measured intervals. **c** Quantification of widths of the superstructure interlayer distances and those of the singlet microtubules in (**b**). n = 55 structures. **d** TEM images of younger centrioles in G1 phase RPE-1 cells transfected with siControl or siHYLS1. Scale bar: 100 nm. n = 7 (siControl) or 12 (siHYLS1) cells. **e** Representative IF images of RPE-1 cells subjected to centriole stability assay with STLC. Before the STLC treatment, the cells were transfected with siControl or siHYLS1. Scale bar: 5 µm. Arrowheads: Centrin foci (centrioles). **f** Quantification of frequency of mitotic cells with <4 Centrin foci in **e**. Only groups treated with STLC for 18 h are shown. n = 3 independent experiments, 50 cells each. **g** IF images of RPE-1 C-Nap1 KO cells subjected to microtubule regrowth assay (30°C, 5 seconds). The cells were transfected with siControl or

siHYLS1 before cold-induced microtubule depolymerization. Scale bars: 10 µm and 1 µm. Arrowheads: magnified areas with older (white) or younger (orange) centrosomes. **h** Quantification of average EB1 intensity around younger centrosomes in (**g**). A.U.: arbitrary unit. n = 150 cells pooled from 3 independent experiments, 50 cells each. **i** IF images of mitotic RPE-1 cells treated with siControl or siHYLS1 for 72 h. Scale bar: 10 µm. Arrowheads: Centrin foci (centrioles). **j** Quantification of frequency of mitotic cells with abnormal mitotic spindles, such as monopolar spindles, asymmetric bipolar spindles with non-centrosomal poles, and bipolar spindles with precociously disengaged centrioles, in (**i**). n = 3 independent experiments, 31, 32, and 33 cells (siControl) or 31, 31, and 32 cells (siHYLS1). **k** IF images of RPE-1 cells transfected with siControl or siHYLS1 and then serum starved for 48 h. Scale bar: 5 µm. **l** Quantification of frequency of interphase cells with primary cilia in (**k**). n = 3 independent experiments, 50 cells each. **m** Quantification of length of primary cilia in (**k**). n = 50 (siControl) or 39 (siHYLS1) cells pooled from 3 independent experiments. **n** Schematic showing the phenotypes resulting from HYLS1 depletion in centrioles, centrosomes, and primary cilia. MT: microtubule. Data are represented as mean ± s.d. P values were calculated by Mann−Whitney U test (**c**, **h**, **m**), one-way ANOVA with Tukey's multiple comparisons test (**f**), or two-tailed unpaired Student's t-test (**j**, **l**). Source data are provided as a Source Data file.

To confirm the inhibitory role of the TUBB CTT in incomplete microtubule assembly in vivo, we overexpressed a TUBB mutant lacking the CTT (TUBBΔCTT; 1-429 a.a.) in RPE-1 cells. Overexpression of TUBBΔCTT, but not of the full-length TUBB (1-444 a.a.), was sufficient to induce the assembly of tubulin-based superstructures without HYLS1 overexpression (Fig. 6d, e and Supplementary Fig. 11a). The superstructures generated upon TUBBΔCTT overexpression were shorter than the HYLS1-induced superstructures, but showed similar cross section architecture and diameters (Supplementary Fig. 11b–d). Furthermore, they exhibited a high stability similar to the HYLS1-induced superstructures and centrioles (Supplementary Fig. 11e–h). These data demonstrate that suppressing the inhibitory role of the TUBB CTT is sufficient for the assembly of incomplete MT superstructures in human cells.

### HYLS1 regulates the C-terminal tail of TUBB to promote the assembly of incomplete microtubules and stable centrioles

A CTT-truncated version of another major β-tubulin isotype TUBB4B (1-429 a.a.), but not of the major α-tubulin isotype TUBA1B (1-438 a.a.), was also able to induce the assembly of similar superstructures when overexpressed (Supplementary Fig. 12a–d). These results suggest that the β-tubulin body without the CTT, but not the α-tubulin body lacking the CTT, has the capacity to assemble these superstructures. However, given that beside TUBB, none of the other β-tubulin isotypes exhibited strong interactions with HYLS1 in the AlphaFold-Multimer predictions (Fig. 4c) and they were dispensable for the HYLS1-dependent superstructure formation (Fig. 4f), it seems likely that the TUBB body is specifically required for the HYLS1-dependent assembly of superstructures. To test this assumption, we replaced the body of TUBB with that of TUBB4B and evaluated the functionality of this swapped protein in this context (Supplementary Fig. 13a–c). Upon siTUBB treatment, RNAi-resistant wildtype TUBB partially reversed the reduction in cells exhibiting HYLS1-induced superstructures, whereas the swapped protein did not (Supplementary Fig. 14a, b). In addition to the body swapping, the replacement of the CTT of TUBB with that of TUBB4B also resulted in a loss of this specific functionality (Supplementary Fig. 14a, b). These results indicate that both the body and the CTT of TUBB are crucial for the HYLS1-dependent assembly of incomplete microtubules.

As shown in Fig. 6a, the structural models show the closest proximity between HYLS1 D211 and the glycine at position 437 (G437) in the TUBB CTT. The distance between the two residues is about 2.7 Å, consistent with that of two atoms forming hydrogen bonds. Most of the other β-tubulin isotypes, including TUBB4B, encode a glutamic

acid (E) instead of G in the corresponding position of their CTTs (Fig. 6b), which could explain the tubulin isotype specificity in the assembly of incomplete MT superstructures (Fig. 4e, f) and stable centrioles (Fig. 5a, b). To investigate the importance of the predicted physical modulation of the TUBB CTT by HYLS1 D211, we substituted TUBB G437 to E which is expected to result in electrostatic repulsion of the HYLS1 D211 residue, consequently interrupting the physical interaction of both proteins. As anticipated, upon siTUBB treatment, the G437E mutant of TUBB did not increase the ratio of cells displaying HYLS1-induced superstructures (Fig. 6f, g). Notably, expression of the G437E mutant of TUBB showed a dominant negative effect on the assembly of incomplete MT superstructures in cells treated with siControl (Supplementary Fig. 15a, b). These data indicate that HYLS1 promotes the assembly of incomplete microtubules through the regulation of the TUBB CTT via its D211 residue.

To determine whether the regulation of the TUBB CTT by HYLS1 is crucial for the structural integrity of centrioles, we assessed their stability by the STLC and cold treatment assays. In both assays, unlike the wildtype protein, the G437E mutant of TUBB failed to stabilize centrioles (Fig. 6h, i and Supplementary Fig. 15c−e). Furthermore, while controlled expression of the wildtype TUBB could rescue the defects in primary ciliogenesis induced upon siTUBB treatment, the G437E mutant of TUBB failed to do so (Supplementary Fig. 15f−h). Thus, these data indicate that the physical modulation of the TUBB CTT by HYLS1 through the HYLS1 D211-TUBB G437 interaction promotes the assembly of incomplete tubules within centriole triplet microtubules.

## Discussion

The molecular mechanisms of the assembly of centriole triplet microtubules with two atypical incomplete microtubules represent a long-standing open question in biology. Since 1984, the presence of regulators that initiate the assembly of incomplete microtubules via regulation of the tubulin C-terminal tail (CTT) has been suggested[11,12]. Here, we identified the ciliopathy protein HYLS1 as the missing regulator, which modulates the CTT of the β-tubulin isotype TUBB to promote the assembly of centriole triplet microtubules in human cells.

In the 1980s study, artificial removal of the tubulin CTTs enabled in vitro reconstitution of incomplete tubule chains resembling the centriole triplet microtubules[11]. A recent in silico analysis further demonstrated that the wobbling CTTs, protruding from the surface of an assembled A-tubule, inhibit the attachment of additional protofilaments to its lateral surface, thereby suppressing the assembly of incomplete microtubules[12]. In this study, we revealed that a TUBB mutant lacking the CTT (TUBBΔCTT) assembles superstructures with

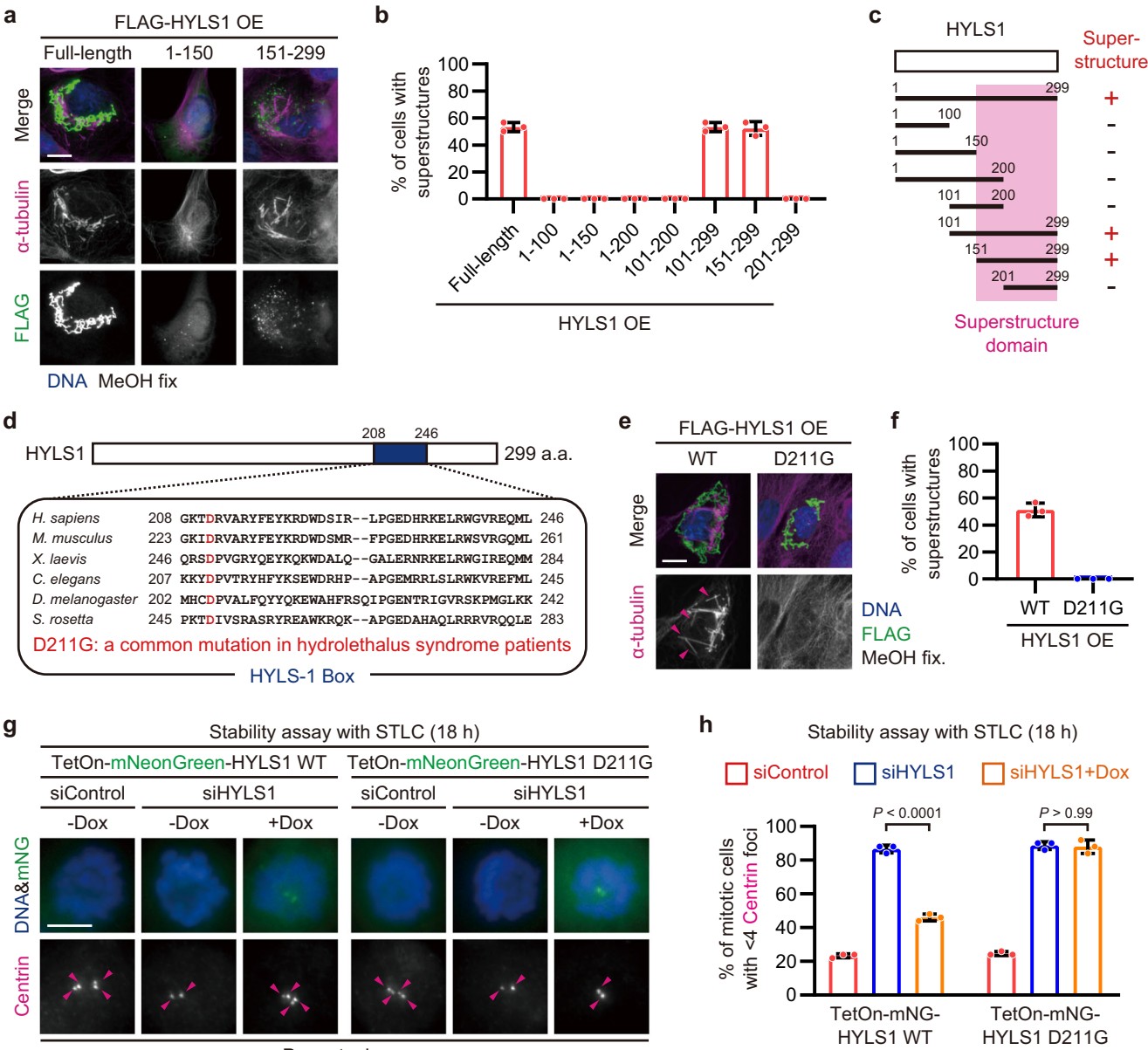

**Fig. 3 | The ciliopathy-related D211G mutant of HYLS1 fails to assemble incomplete microtubules and stable centrioles. a** Representative IF images of RPE-1 cells transfected with the indicated constructs. Scale bar: 10 μm. **b** Quantification of frequency of interphase cells with incomplete MT superstructures in (**a**). n = 3 independent experiments, 30 cells each. **c** Schematic showing a specific region of HYLS1 that is required for the assembly of incomplete MT superstructures (superstructure domain). **d** Schematic showing amino acid sequences of HYLS-1 Box in the indicated species. **e** IF images of RPE-1 cells transfected with pCMV-FLAG-HYLS1 WT or pCMV-FLAG-HYLS1 D211G. Scale bar: 10 μm. Arrowheads: incomplete MT superstructures. **f** Quantification of frequency of

interphase cells with incomplete MT superstructures in (**e**). n = 3 independent experiments, 30 cells each. **g** Representative IF images of RPE-1 TetOn-mNeonGreen-X cells (X = HYLS1 WT or D211G) subjected to centriole stability assay with STLC. Before the STLC treatment, the cells were transfected with siControl or siHYLS1 and were treated with doxycycline. Scale bar: 5 μm. Arrowheads: Centrin foci (centrioles). **h** Quantification of frequency of mitotic cells with <4 Centrin foci in (**g**). Only groups treated with STLC for 18 h are shown. n = 3 independent experiments, 50 cells each. Data are represented as mean ± s.d. P values were calculated by one-way ANOVA with Tukey's multiple comparisons test (**h**). Source data are provided as a Source Data file.

incomplete microtubules in human cells (Fig. 6d, e), demonstrating the inhibitory role of the β-tubulin CTT in the formation of incomplete microtubules in cells. Furthermore, we discovered that the ciliopathy protein HYLS1 interacts with the CTT of TUBB (Fig. 6a) to generate superstructures with incomplete microtubules in human cells (Fig. 1g). In the AlphaFold-based structural models, HYLS1 limited the potential positions occupied by the TUBB CTT (Fig. 6c), indicating that HYLS1 may fold or anchor the wobbling CTT of TUBB. Based on these structural predictions and experimental observations, we propose that, instead of undergoing cleavage, the CTT of TUBB is physically sequestered by HYLS1 to promote the assembly of incomplete tubules within centriole triplet microtubules.

The triplet microtubule consists of two sequential incomplete tubules, B- and C-tubules. Which of the two incomplete tubules does HYLS1 assemble? The HYLS1-induced superstructures show a sequential pattern of an incomplete microtubule attached to the lateral surface of adjacent incomplete tubule (Fig. 2a). This pattern is similar to the attachment of C-tubule to the incomplete B-tubule within

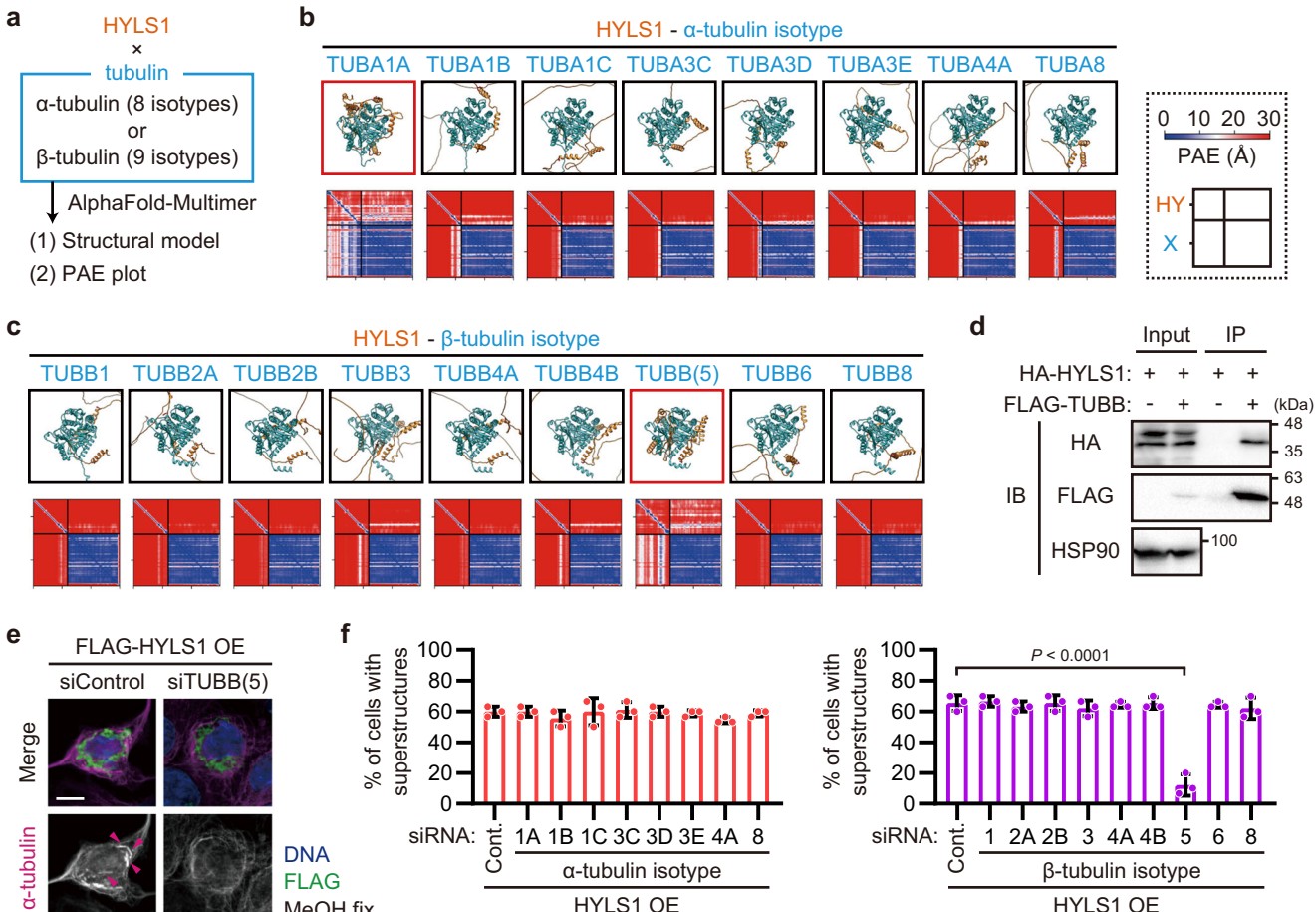

**Fig. 4 | The β-tubulin isotype TUBB is specifically required for the HYLS1-dependent assembly of incomplete microtubules. a** Schematic of AlphaFold-based screening. PAE: predicted aligned error. **b, c** Structural model and PAE plot of the protein complex of HYLS1 and each α-tubulin (**b**) or β-tubulin (**c**) isotype generated by AlphaFold-Multimer. **d** Immunoblotting images of immunoprecipitated lysates from HEK293 cells transfected with the indicated constructs. Immunoprecipitation was performed using an antibody against FLAG-tag. IP: immunoprecipitation, IB: immunoblotting. **e** Representative IF images of RPE-1 cells transfected with pCMV-FLAG-HYLS1 and the indicated siRNAs. For each group, two distinct siRNAs were mixed and used, except for siTUBA3D. Scale bar: 10 μm. Arrowheads: incomplete MT superstructures. **f** Quantification of frequency of interphase cells with incomplete MT superstructures in (**e**). Cont.: Control. $n = 3$ independent experiments, 30 cells each. Data are represented as mean ± s.d. $P$ value was calculated by one-way ANOVA with Dunnett's multiple comparisons test (**f**). Source data are provided as a Source Data file.

centrioles, indicating that HYLS1 promotes the nucleation of C-tubule. Nevertheless, HYLS1 would also promote the nucleation of B-tubule from the complete A-tubule since HYLS1-depleted cells exhibited aberrant centrioles lacking both B- and C-tubules (Fig. 2d). In support of this, in *C. elegans*, HYLS1, which localizes just outside the singlet A-tubules, is crucial for the assembly of B-tubules in cilia[24,40]. Taken together, HYLS1 likely initiates the assembly of both B- and C-tubules within centrioles by physically modulating the TUBB CTTs of A- and B-tubules, respectively (Fig. 6j).

The β-tubulin isotype TUBB is specifically required for the HYLS1-dependent superstructure formation (Fig. 4e, f). However, a CTT truncated version of another β-tubulin isotype TUBB4B was also able to induce the assembly of similar superstructures (Supplementary Fig. 12c, d), raising the possibility that there are additional regulators modulating the CTTs of different tubulin isotypes to promote incomplete microtubule formation. Notably, the evolutionary conservation of HYLS1 has only been confirmed up to the level of ciliates[23]. Given this superficially limited phylogenetic distribution compared to the widespread conservation of incomplete microtubules within centrioles across eukaryotes[4], alternative pathways might support incomplete microtubule assembly in species lacking HYLS1. This study should encourage subsequent research to seek for additional proteins

and mechanisms involved in the assembly of incomplete microtubules in diverse organisms.

Live-cell imaging of the assembly process of the superstructures demonstrated that the formation of the HYLS1 structures slightly preceded that of the tubulin-based superstructures (Fig. 1h). This precedence of HYLS1 structures was also captured by STED microscopy (Fig. 1i, Side-view). These observations suggest that HYLS1 may further support the stacking of tubulin-based superstructures, resulting in their elongation. Indeed, although overexpressed TUBBΔCTT solely assembled tubulin-based superstructures in cells, these were shorter than those detected in HYLS1 overexpressed cells (Supplementary Fig. 11d). Therefore, HYLS1 may also promote the stacking of the centriole triplet microtubule in addition to its assembly.

How do centriole triplet microtubules avoid the formation of additional incomplete D-tubules? Recent ultrastructure expansion microscopy data show that C-tubules specifically undergo polyglutamylation, a process characterized by additions of multiple glutamate (E) residues within the tubulin CTT[41]. The CTT of TUBB is modulated by the D211 residue of HYLS1 (Fig. 6a, c) to promote the assembly of incomplete microtubules (Fig. 6f, g). The polyglutamylation of C-tubules therefore has the potential to interrupt the interaction with the D211 residue of HYLS1 due to the electrostatic repulsion

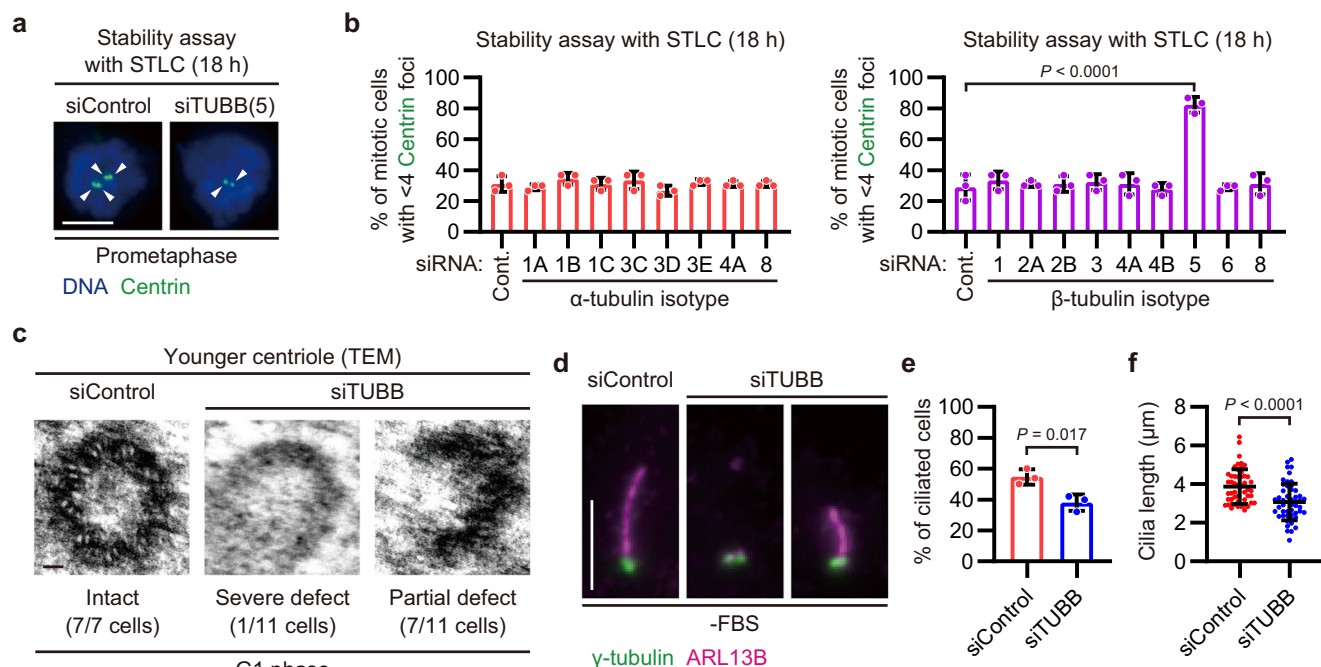

**Fig. 5 | TUBB provides the structural integrity of centriole triplet microtubules.**
**a** Representative IF images of RPE-1 cells subjected to centriole stability assay with STLC. Before the STLC treatment, the cells were transfected with the indicated siRNAs. For each group, two distinct siRNAs were mixed and used, except for siTUBA3D. Scale bar: 5 μm. Arrowheads: Centrin foci (centrioles). **b** Quantification of frequency of mitotic cells with <4 Centrin foci in (**a**). Only groups treated with STLC for 18 h are shown. Cont.: Control. *n* = 3 independent experiments, 30 cells each. **c** TEM images of younger centrioles in G1 phase RPE-1 cells transfected with siControl or siTUBB. Scale bar: 100 nm. *n* = 7 (siControl) or 11 (siTUBB) cells. **d** IF images of RPE-1 cells transfected with siControl or siTUBB and then serum starved for 48 h. Scale bar: 5 μm. **e** Quantification of frequency of interphase cells with primary cilia in (**d**). *n* = 3 independent experiments, 50 cells each. **f** Quantification of length of primary cilia in (**d**). *n* = 50 cells pooled from 3 independent experiments. Data are represented as mean ± s.d. *P* values were calculated by one-way ANOVA with Dunnett's multiple comparisons test (**b**), two-tailed unpaired Student's *t*-test (**e**), or Mann–Whitney *U* test (**f**). Source data are provided as a Source Data file.

occurring between the negatively charged residues. Consequently, the polyglutamylated CTTs of C-tubules may be left in a state of wobbling even in the presence of HYLS1, thereby preventing the attachment of any additional incomplete tubule protofilaments. Consistent with this hypothesis, the HYLS1-induced superstructures did not exhibit polyglutamylation (Supplementary Fig. 3a) and displayed continuous incomplete microtubule chains (Fig. 2a). Therefore, the C-tubule-specific polyglutamylation may inhibit the HYLS1-depedent regulation of the TUBB CTT to define the number of incomplete tubules within centriole triplet microtubules.

The centriole exhibits a high stability for a microtubule-based structure[42], which ensures its pivotal roles as the core structures of centrosomes, cilia, and flagella[8–10]. Notably, centriole microtubules persist under conditions that cause immediate depolymerization of cytoplasmic microtubules[27,28]. However, the mechanisms that provide the centriole triplet microtubules with resistance to depolymerization have remained elusive. Here we show that the HYLS1-induced superstructures display similarly high stability (Fig. 1k–m and Supplementary Fig. 3d–f) even though they are composed of microtubules and do not recruit centriolar proteins (Supplementary Fig. 3b, c), suggesting that this specific microtubule arrangement with incomplete tubules confers the resistance to depolymerization. In support of this hypothesis, the superstructures generated upon TUBBΔCTT overexpression also exhibited a high stability independently of HYLS1 (Supplementary Fig. 11e–h). How could this effect be explained? Each incomplete microtubule of the HYLS1-induced superstructures is tethered to the two neighboring tubules through unique lateral interactions, which are also observed at the outer junctions within centriole triplet microtubules[5–7]. This is a two-to-one protofilament interaction and therefore likely more stable than the canonical lateral

interaction within a single microtubule that is one-to-one protofilament[43]. Thus, these unique lateral interactions at the outer junctions may contribute to the extraordinary stability of centrioles, in addition to other reinforcing factors such as A-C linkers and inner scaffolds[5–7,32,44,45].

Overall, our findings provide fundamental insights into the unique assembly mechanisms and properties of the centriole triplet microtubule, a nano-scale cellular architecture conserved for over a billion years.

## Methods

### Cell culture
RPE-1 (ATCC, CRL-4000), HeLa (ECACC, 93021013), HEK293 (ECACC, 85120602), and HEK GP2-293 (Clontech, 631512) cells were authenticated by the suppliers via short tandem repeat profiling before purchase. It was confirmed that the cells were not contaminated with mycoplasma using TaKaRa PCR Mycoplasma Detection Set (TaKaRa, 6601) at least once a year (most recently in July, 2023). The HeLa, HEK293, and HEK GP2-293 cells were cultured in Dulbecco's modified Eagle's medium (DMEM; Nacalai Tesque, 08459-64) supplemented with 10% fetal bovine serum (FBS; NICHIREI, 175012), 100 U/ml penicillin, and 100 μg/ml streptomycin (Nacalai Tesque, 09367-34) at 37°C in a 5% CO2 atmosphere. The RPE-1 cells were cultured in DMEM/F-12 medium (Nacalai Tesque, 11581-15) supplemented with 10% FBS, 100 U/ ml penicillin, and 100 μg/ml streptomycin, at 37°C in a 5% CO2 atmosphere.

### Drug treatment
The following chemicals were used in the cell biological experiments: STLC (Sigma, 164739), nocodazole (Wako, 31430-18-9), doxycycline

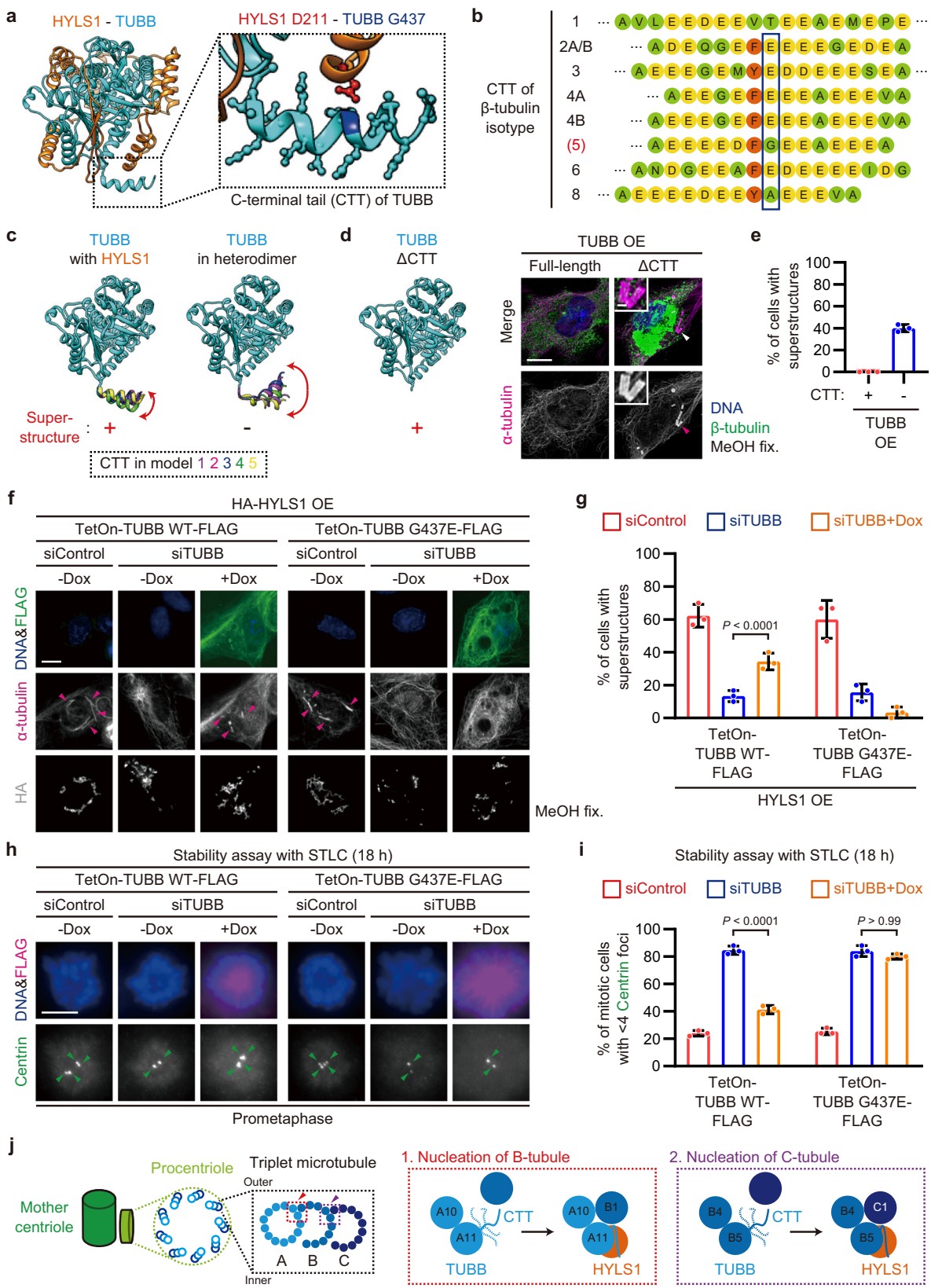

(Sigma, D9891), EdU (Thermo Fisher Scientific, C10640), puromycin (InvivoGen, ant-pr-5), hygromycin B Gold (InvivoGen, ant-hg-1). To induce prometaphase arrest, cells were treated with 10 μM STLC. To depolymerize cytoplasmic microtubules, cells were treated with 10 μM nocodazole. To induce the TetOn system-dependent protein expression, cells were treated with 1 μg/ml docycycline. To determine the cell cycle phase, cells were treated with 10 μM EdU for 30–60 min, then fixed and stained using Click-iT Plus EdU Alexa Flour 647 imaging kit (Thermo Fisher Scientific, C10640) according to the manufacturer's instruction. To isolate cells expressing drug-resistant sequences, the cells were treated with puromycin (5 μg/ml) or hygromycin B Gold (100 μg/ml) for 1 day or 3 days, respectively.

**Fig. 6 | HYLS1 regulates the C-terminal tail of TUBB to promote the assembly of incomplete microtubules and stable centrioles. a** Structural model of HYLS1-TUBB complex generated by AlphaFold-Multimer. C-terminal tail (CTT) of TUBB is magnified. Orange: HYLS1, red: HYLS1 D211, cyan: TUBB, blue: TUBB G437. **b** Schematic of the CTT amino acid sequences of β-tubulin isotypes. Orange: aromatic, light green: uncharged, yellow: negatively charged. **c** Structural models of TUBB either in HYLS1-TUBB complex or TUBB-TUBA1B complex generated by AlphaFold-Multimer. For each complex, five models were overlaid. **d** Structural model of TUBB lacking the CTT generated by AlphaFold-Multimer and IF images of RPE-1 cells transfected with pCMV-TUBB with or without the CTT. Scale bars: 10 μm and 1 μm. Arrowheads: magnified area. **e** Quantification of frequency of interphase cells with tubulin-based superstructures similar to incomplete MT superstructures in (**d**). $n = 3$ independent experiments, 30 cells each. **f** Representative IF images of RPE-1 TetOn-X-FLAG cells (X = TUBB WT or G437E) transfected with pCMV-HA-

HYLS1 and siControl or siTUBB, and treated with doxycycline. Scale bar: 10 μm. Arrowheads: incomplete MT superstructures. **g** Quantification of frequency of interphase cells with incomplete MT superstructures in (**f**). $n = 3$ independent experiments, 30 cells each. **h** Representative IF images of RPE-1 TetOn-X-FLAG cells (X = TUBB WT or G437E) subjected to centriole stability assay with STLC. Before the STLC treatment, the cells were transfected with siControl or siTUBB and were treated with doxycycline. Scale bar: 5 μm. Arrowheads: Centrin foci (centrioles). **i** Quantification of frequency of mitotic cells with <4 Centrin foci in (**h**). Only groups treated with STLC for 18 h are shown. $n = 3$ independent experiments, 50 cells each. **j** Schematic showing a speculative model of centriole triplet microtubule assembly promoted by HYLS1. Data are represented as mean ± s.d. $P$ values were calculated by one-way ANOVA with Tukey's multiple comparisons test (**g**, **i**). Source data are provided as a Source Data file.

## Antibodies

The following primary antibodies were used: rabbit polyclonal antibodies against RFP/mCherry [MBL, PM005, Lot 046, immunofluorescence (IF) 1:500, immunoblotting (IB) 1:1000], CEP97 [Novus Biologicals, NBP1-83591, Lot A107064, IF 1:500], α-tubulin [MBL, PM054, Lot 007, IF 1:500, IB 1:1000], β-tubulin [Thermo Fisher Scientific, PA5-16863, Lot XG3654589, IF 1:500], acetylated tubulin [Abcam, ab179484, IF 1:500], FLAG [Sigma, F7425, Lot 086M4803V, IF 1:500], HA [Abcam, ab9110, IF 1:1000], PCNT [Abcam, ab4448, IF 1:1000], HYLS1 [Novus Biologicals, NBP1-56899, Lot QC25211-90821, IB 1:1000], CEP152 [Bethyl Laboratories, A302-480A, IF 1:1000], CEP192 [Bethyl Laboratories, A302-324A, IF 1:1000], CP110 [Proteintech, 12780-1-AP, IF 1:500], ANKRD26 [GeneTex, GTX128255, Lot 41458, IF 1:1000], CEP44 [Proteintech, 24457-1-AP, IF 1:500], CEP57 [GeneTex, GTX115931, Lot 40289, IF 1:500], CEP63 [Proteintech, 16268-1-AP, IF 1:500], CEP135 [Abcam, ab196809, Lot GR199116-6, IF 1:500], CEP295 [Sigma, HPA038596, Lot R33456, IF 1:500], CPAP [Proteintech, 11517-1-AP, IF 1:500], PLK1 [Bethyl Laboratories, A300-251A, IF 1:500], and ARL13B [Proteintech, 17711-1-AP, IF 1:500]; goat polyclonal antibody against PCNT [Santa Cruz Biotechnology, sc-28145, Lot C1716, IF 1:500]; mouse monoclonal antibodies against Centrin [Merck Millipore, 04-1624, Clone 20H5, IF 1:1000], γ-tubulin [Sigma, T5326, Clone GTU88, IF 1:1000], α-tubulin [Sigma, T5168, Clone B-5-1-2, IF 1:1000], acetylated tubulin [Sigma, T7451, Clone 6-11B-1, IF 1:500], polyglutamylation modification [AdipoGen, AG-20B-0020-C100, Clone GT335, IF 1:500], monoglycylated tubulin [Merck Millipore, MABS277, Clone TAP952, IF 1:500], polyglycylated tubulin [Merck Millipore, MABS276, Clone AXO49, IF 1:500], FLAG [Sigma, F1804, Clone M2, IF 1:1000, IB 1:1000], HA [Biolegend, 901501, Clone 16B12, IF 1:500], SAS6 [Santa Cruz Biotechnology, sc-81431, IF 1:500], HSP90 [BD Biosciences, 610419, IB 1:2000], EB1 [BD Biosciences, 610534, Clone 5/EB1, IF 1:500]; rat monoclonal antibodies against HA [Merck, 11867423001, Clone 3F10, IF 1:500], Centrin 2 [BioLegend, 698602, Clone W16110A, IF 1:500]. The following secondary antibodies were used: Alexa Fluor 488 donkey anti-mouse IgG (H + L) (Thermo Fisher Scientific, A32766, IF 1:500), Alexa Fluor 488 donkey anti-rabbit IgG (H + L) (Thermo Fisher Scientific, A32790, IF 1:500), Alexa Fluor 488 donkey anti-rat IgG (H + L) (Thermo Fisher Scientific, A48269, IF 1:500), Alexa Fluor 555 donkey anti-mouse IgG (H + L) (Thermo Fisher Scientific, A32773, IF 1:500), Alexa Fluor 555 donkey anti-rabbit IgG (H + L) (Thermo Fisher Scientific, A32794, IF 1:500), Alexa Fluor 647 donkey anti-mouse IgG (H + L) (Thermo Fisher Scientific, A32787, IF 1:500), Alexa Fluor 647 donkey anti-rabbit IgG (H + L) (Thermo Fisher Scientific, A32795, IF 1:500), Alexa Fluor 647 donkey anti-goat IgG (H + L) (Thermo Fisher Scientific, A32849, IF 1:500), Alexa Fluor 647 donkey anti-rat IgG (H + L) (Thermo Fisher Scientific, A48272, IF 1:500), horseradish peroxidase-conjugated goat polyclonal antibodies against mouse IgG (Promega, W4021, IB 1:5000), and horseradish peroxidase-conjugated goat polyclonal antibodies against rabbit IgG (Promega, W4011, IB 1:5000).

## Plasmid construction

Full-length HYLS1 cDNA (NCBI NP_001128265.1) was amplified from a cDNA library of RPE-1 cells. Mutant constructs of HYLS1, including D211G and RNAi-resistant variants, were generated using PrimeSTAR mutagenesis basal kit (TaKaRa, R046A). Truncated versions of HYLS1 were amplified from the full-length HYLS1 cDNA. These constructs were cloned into pCMV5, pCMV5-FLAG, pCMV5-HA, or pRetroX-TRE3G-mNeonGreen[46,47]. Full-length mScarlet-i cDNA was amplified from pQPXIP-mScarlet-i-centrin2[46] and subsequently cloned into pCMV5-HYLS1. Full-length TUBB cDNA (NCBI NP_821133.1) was amplified from a cDNA library of RPE-1 cells and subsequently cloned into pCMV5, pCMV5-FLAG, pQCXIZ-mNeonGreen, or pRetroX-TRE3G-FLAG. RNAi-resistant TUBB cDNA was generated by a two-step PCR and following a ligation reaction using Ligation-convenience kit (NIPPON GENE, 315-05963). TUBB G437E cDNA was generated using PrimeSTAR mutagenesis basal kit from the full-length RNAi-resistant TUBB cDNA. TUBB cDNA lacking the C-terminal tails (CTT) was amplified from the full-length TUBB cDNA. These constructs were cloned into pCMV5, pCMV5-FLAG, or pRetroX-TRE3G-FLAG. Full-length TUBA1B cDNA (NCBI NP_006073.2) was amplified from a cDNA library of RPE-1 cells. TUBA1B cDNA lacking the CTT was amplified from the full-length TUBA1B cDNA. These constructs were cloned into pCMV5. Full-length TUBB4B cDNA (NCBI NP_006079.1) was amplified from a cDNA library of RPE-1 cells and subsequently cloned into pCMV5, pCMV5-FLAG, or pRetroX-TRE3G-FLAG. TUBB4B cDNA lacking the CTT was amplified from the full-length TUBB4B cDNA and cloned into pCMV5 and pRetroX-TRE3G-TUBB CTT-FLAG. The TUBB cDNA lacking the CTT described above was cloned into pRetroX-TRE3G-TUBB4B CTT-FLAG. HYLS1 homologs in *C.elegans* and *D.melanogaster* cDNAs were generated by IDT gBlocks Gene Fragments service and subsequently cloned into pCMV5-FLAG. GuideRNA oligo targeting the N-terminus of HYLS1 was annealed and inserted into the *Bbs*I site of pX330-U6-Chimeric_BB-CBh-hSpCas9 (pX330-hSpCas9; Addgene, 42230). Homology arms of the HYLS1 locus were amplified from the genomic DNA of HeLa cells and then cloned into pBluescript using In-Fusion HD Cloning Kit (Takara, 639650). Subsequently, mCherry cassette containing a hygromycin-resistance gene was cloned into the middle of homology arms. These two plasmids were named as pX330-hSpCas9-sgHYLS1 and pBluescript-HygR-mCherry-HYLS1 N ter., respectively. Unless otherwise noted, construct amplification was carried out using KOD One PCR Master Mix (TOYOBO, KMM-201), and cloning of the constructs into vectors was performed using In-Fusion HD Cloning Kit or NEBuilder HiFi Assembly Master Mix (NEW ENGLAND BioLabs, E2621S). The primer sequences are listed in Supplementary Table 1.

## Plasmid transfection

Plasmid DNA transfection into HeLa or RPE-1 cells was performed using Lipofectamin 2000 (Thermo Fisher Scientific, 11668-019) or TransIT-X2 reagent (Mirus, MIR 6003) according to the

manufacturer's instructions. Plasmid DNA transfection into HEK293 or HEK GP2-293 cells was performed using PEI MAX (Polysciences, 24765-1) according to the manufacturer's instruction. Unless otherwise noted, plasmid DNA transfection was performed 24 h before cell collection or fixation.

## RNA interference

The following Silencer Select siRNAs (Thermo Fisher Scientific) were used: siHYLS1#1 (s47731), siHYLS1#2 (s47732), siTUBA1A#1 (s15400), siTUBA1A#2 (s15401), siTUBA1B#1 (s20288), siTUBA1B#2 (s20289), siTUBA1C#1 (s195488), siTUBA1C#2 (s224963), siTUBA3C#1 (s194842), siTUBA3C#2 (s194843), siTUBA3D (s195595), siTUBA3E#1 (s41356), siTUBA3E#2 (s195587), siTUBA4A#1 (s14497), siTUBA4A#2 (s14498), siTUBA8#1 (s28671), siTUBA8#2 (s28672), siTUBB1#1 (s37479), siTUBB1#2 (s37480), siTUBB2A#1 (s14500), siTUBB2A#2 (s14501), siTUBB2B#1 (s51294), siTUBB2B#2 (s223916), siTUBB3#1 (s20296), siTUBB3#2 (s20297), siTUBB4A#1 (s20299), siTUBB4A#2 (s20300), siTUBB4B#1 (s20301), siTUBB4B#2 (s20302), siTUBB#1 (s284), siTUBB#2 (s283), siTUBB6#1 (s39180), siTUBB6#2 (s39182), siTUBB8#1 (s51282), siTUBB8#2 (s196882), and siControl (4390843, negative control). siRNAs lacking numbering in figures refer to #1. We could obtain only one specific siTUBA3D from the supplier. Transfection of siRNA was performed using Lipofectamin RNAiMAX (Thermo Fisher Scientific, 13778-150) according to the manufacturer's instruction. Unless otherwise noted, transfection of siRNA was performed 48 h before cell collection or fixation. The siRNA sequences are listed in Supplementary Table 2.

## Generation of a cell line by CRISPR-mediated knock-in

HeLa cells expressing endogenously tagged mCherry-HYLS1 (HeLa mCherry-HYLS1) were established using the CRISPR-Cas9 system[48,49]. Both pX330-hSpCas9-sgHYLS1 and pBluescript-HygR-mCherry-HYLS1 N ter. were introduced into HeLa cells. Positive clones were subsequently isolated using the limited dilution method with hygromycin B Gold.

## Generation of cell lines by retroviral-mediated integration

RPE-1 cells stably expressing Tet3G transactivator (RPE-1 Tet3G cells) were previously described[46]. RPE-1 Tet-On cell lines were established by retroviral-mediated integration. Each pRetroX-TRE3G plasmid and pCMV-VSV-G (Addgene, 8454) were transfected into HEK GP2-293 cells. The medium was harvested and filtered through a 0.45 µm filter (Merck Millipore, SLHVR33RS). Pre-seeded RPE-1 Tet3G cells were infected with the virus-containing medium, supplemented with fresh medium, FBS, and 4 µg/ml polybrene (Nacalai Tesque, 12996-81). RPE-1 cells stably expressing TUBB-mNeonGreen were also established by retroviral-mediated integration. pQCXIZ-TUBB-mNeonGreen and pCMV-VSV-G were transfected into HEK GP2-293 cells. The medium was harvested and filtered through a 0.45 µm filter. Pre-seeded RPE-1 cells were infected with the virus-containing medium, supplemented with fresh medium, FBS, and 4 µg/ml polybrene.

## Genomic PCR

Genomic DNA was extracted and purified from cells using NucleoSpin DNA RapidLyse kit (Macherey-Nagel, 740100.50). The region of interest was amplified by PCR using KOD One PCR Master Mix and specific primers. Subsequently, the PCR products were analyzed by 1% agarose gel electrophoresis. The primer sequences are listed in Supplementary Table 3.

## STLC treatment assay

Cells were treated with STLC for 1 h or 18 h before fixation. Based on the chromosome shapes, prometaphase cells were selected for analyses.

## Cold treatment assay

Cells were treated with EdU for 30 min and were then incubated on ice for 1 h before fixation. Based on the signals of EdU and the numbers of Centrin foci (centrioles), G1 phase cells were selected for analyses.

## Microtubule regrowth assay

Cells were incubated on ice for 1 h to depolymerize cytoplasmic microtubules. The cells were then incubated at 30°C for 5 seconds and quickly fixed. To distinguish between the microtubules from two centrosomes, RPE-1 C-Nap1 KO cells[46] which lack centrosome linkers were used. The younger centrioles were distinguished by the absence of ANKRD26 foci (distal appendages).

## Cytoplasmic extraction

Cells were incubated on ice for 1 h to depolymerize cytoplasmic microtubules and then treated with CSK buffer [10 mM PIPES pH 6.8, 100 mM NaCl, 300 mM sucrose, 3 mM MgCl$_2$, and 1 mM EGTA] with 0.5% Triton X-100 to remove their cytoplasmic fraction.

## Fluorescence live-cell imaging

RPE-1 cells stably expressing TUBB-mNeonGreen were cultured in 35-mm glass-bottom dishes (Greiner-bio-one, 627870) at 37°C in a 5% CO2 atmosphere. 12 h before imaging, the cells were transfected with pCMV-mScarlet-i-HYLS1. Time-lapse imaging was carried out with a spinning disk confocal scanner box, the Cell Voyager CV1000 (Yokogawa Electric Corp) equipped with a 40× oil-immersion objective and a back-illuminated EMCCD camera. Z-stack images were captured every 5 min for 24 h using a X40 1.30 NA oil-immersion lens.

## Immunofluorescence (IF)

Cells seeded on coverslips (Matsunami, No. 1 or No. 1 s) were fixed with ice-cold methanol for 10 min at −30°C or 4% paraformaldehyde for 20 min at room temperature. After washing three times with PBS, the cells were permeabilized by PBS with 0.05% Triton X-100 (PBS-X) for 5 min and subsequently blocked by 1% BSA in PBS-X for 30 min at room temperature. The cells were then incubated with primary antibodies overnight at 4°C and were then washed three times with PBS-X, followed by incubation with secondary antibodies and Hoechst 33258 solution (Nacalai Tesque, 19173-41; dilution 1:5000) for 1 h at room temperature. The cells were washed three times with PBS-X and mounted onto glass slides (Matsunami, S0318).

## Fluorescence microscopy

For all immunofluorescence (IF) imaging except for Figs. 1f–i, 6d and Supplementary Figs. 2a, 4a, 11a, b, an Axioplan2 fluorescence microscope (Carl Zeiss) with 63× or 100×/1.4 NA plan apochromatic lens objective was used. Unless otherwise noted, the Z interval was set to 250 nm and 21 sections were sequentially obtained for one field. Maximum intensity Z-projections of the representative images were generated using the ImageJ software (version 1.8.0). For imaging shown in Fig. 1f, i and Supplementary Fig. 11b, a Leica TCS SP8 STED 3X system with a Leica HCPL APO 100×/1.40 NA OIL STED WHITE objective and 660 nm gated STED. Scan speed was set to 400 Hz in combination with threefold line average in a 1024 × 256 format (pixel size 14.2 nm). The STED images underwent deconvolution processing using the Lightning system (Leica). For imaging shown in Figs. 1g, 6d and Supplementary Fig. 11a, a 980 laser-scanning microscope with AiryScan2 (Carl Zeiss) with ×63 plan apochromatic lens objective was used. The Confocal images underwent AiryScan processing or LSM plus processing (Leica). The Z interval was set to 140 nm and 11 sections were sequentially obtained for one field. Maximum intensity Z-projections of the representative images were generated using the ImageJ software (version 1.8.0).

### Transmission electron microscopy (TEM)

Cells seeded on coverslips (Matsunami, No. 1) were fixed with 2.5% glutaraldehyde in 0.1 M phosphate buffer for 1 h at room temperature. After washed three times with 0.1 M phosphate buffer and three times with 0.1 M cacodylate-HCl (pH 7.4), the cells were post-fixed with 1% $OsO_4$ in 0.1 M cacodylate-HCl (pH 7.4) for 1 h on ice. The cells were then washed twice with DDW and incubated with 0.5% uranyl acetate (UA) in DDW for 40 min at room temperature in the dark. The cells were rinsed with 70% ethanol and washed with 80%, 90%, 95% (once for each), and 100% (twice) ethanol for 10 min at room temperature. After dehydration with QY-1 (Nisshin EM, 310) for 10 min at room temperature, the cells were rotated in a 1:1 mixture of QY-1 and Durcupan (Sigma, 44610) overnight at room temperature. The cells were then transferred to Durcupan and rotated for 2 h at room temperature twice. Fresh Durcupan was poured into plastic tubes and the cells on coverslips were embedded at 60 °C for 48 h. The coverslips with embedded cells were separated from the tubes by liquid nitrogen. Ultrathin sections (50-60 nm) were prepared with an ultramicrotome (Leica, REICHERT NIS-SEI) and picked up onto formvar-covered meshes (Okenshoji, 09-1029). The meshes were stained with 4% UA for 5 min in the dark under humid conditions at room temperature. After washing with DDW, the meshes were treated with lead citrate solution in the presence of solid KOH for 1 min at room temperature and subsequently washed with DDW. The meshes were imaged using TEM (JEOL, 1200EX) operating at 80-90 kV and a CCD camera (JEOL, Veleta). The younger centrioles were distinguished by the absence of appendages.

### Correlative light and electron microscopy (CLEM)

Cells seeded on 35-mm No. 1.5 glass-bottom gridded dish (MatTek, P35G-1.5-14-CGRD) were pre-fixed with 4% paraformaldehyde in 0.1 M phosphate buffer for 20 min at room temperature. The pre-fixed cells were then pre-observed with BZ-X800 systems (Keyence) to determine the target areas. Subsequently, the cells were fixed with 2.5% glutaraldehyde in 0.1 M phosphate buffer for 1 h at room temperature. After washed three times with 0.1 M phosphate buffer and three times with 0.1 M cacodylate-HCl (pH 7.4), the cells were post-fixed with 1% $OsO_4$ in 0.1 M cacodylate-HCl (pH 7.4) for 1 h on ice. The cells were then washed twice with DDW and the coverslips were removed from the dishes using coverslip removal fluid (MatTek, P-DCF-OS-30). The cells were rinsed with 70% ethanol and washed with 80%, 90%, 95% (once for each), and 100% (twice) ethanol for 10 min at room temperature. The cells were then transferred to aluminum dishes. After dehydration with QY-1 for 10 min at room temperature, the cells were slowly shaken in a 1:1 mixture of QY-1 and Durcupan overnight at room temperature. The cells were then transferred to Durcupan and slowly shaken for 2 h at room temperature twice. Fresh Durcupan was poured into plastic tubes and embedded onto the coverslips in new aluminum dishes at 60 °C for 48 h. The coverslips with the embedded cells were separated from the tubes by liquid nitrogen. Ultrathin sections (50 nm) of the area determined by the pre-observation were prepared by an ultramicrotome and picked up onto formvar-covered meshes. The meshes were then stained with 4% UA for 5 min in the dark under humid conditions at room temperature. After washing with DDW, the meshes were treated with lead citrate solution in the presence of solid KOH for 1 min at room temperature and washed with DDW. The meshes were imaged using TEM (JEOL, 1200EX) operating at 80-90 kV and a CCD camera (JEOL, Veleta).

### Immunoprecipitation (IP)

HEK293 cells were harvested 24 h after transfection and lysed on ice in lysis buffer [50 mM Tris-HCl pH 7.5, 200 mM NaCl, 0.5% Triton X-100, 1 mM DTT, and 1:500 protease inhibitor cocktail (Nacalai Tesque, 25955-11)]. The lysates were centrifuged at 20800 $g$ for 10 min to remove insoluble material. For immunoprecipitation of FLAG-tagged proteins, the lysates were incubated with M2 agarose gel conjugated

with a FLAG antibody (Merck Millipore, A2220) for 2 h at 4 °C. After washed at least three times with the lysis buffer, the agarose beads were resuspended in sodium dodecyl sulfate (SDS) sample buffer (Nacalai Tesque, 09499-14) and heated for 5 min at 95 °C. The supernatant was collected after centrifugation at 1000 $g$ for 2 min.

### Immunoblotting (IB)

Protein samples in SDS sample buffer (Nacalai Tesque, 09499-14) were subjected to SDS-PAGE on a 10-12% polyacrylamide gel and subsequently transferred to Immobilon-P membrane (Merck Millipore, IPVH85R). The membrane was blocked with 5% skimmed milk in PBS containing 0.02% Tween (PBS-T) for 60 min at room temperature and then washed three times with PBS-T. The membrane was incubated with primary antibodies in 5% BSA in PBS-T overnight at 4°C. After washed three times with PBS-T, the membrane was incubated with secondary antibodies in 5% skimmed milk in PBS-T for 1 h at room temperature. The membrane was washed three times with PBS-T again before chemiluminescent detection. Signal detection was carried out with Chemi-Lumi One L (Nacalai Tesque, 07880), Chemi-Lumi One Super (Nacalai Tesque, 02230), or Chemi-Lumi One Ultra (Nacalai Tesque, 11644) and a Chemi Doc XRS+ (BioRad).

### Real-time PCR

Total RNA isolated from cells using TRIzol Reagent (Thermo Fisher Scientific, 15596026) was subjected to reverse transcription with Reverse Transcription Kit (QIAGEN, RT31-020). The resulting cDNA was analyzed by real-time PCR using TB Green Premix Ex Taq II (TaKaRa, RR820A) and specific primers on a LightCycler 480 (Roche). The mRNA levels of these genes were normalized against GAPDH mRNA. The primer sequences are listed in Supplementary Table 4.

### Screening using the cancer dependency map (DepMap)

All the Chronos scores (DepMap Public 21Q3 CRISPR_gene_effect) were downloaded from the DepMap website (October 31st, 2020). The first step was performed with Microsoft Office Excel (version 2401). All the correlation coefficients were calculated and all the registered genes were sorted by the averages of them. At this stage, the top 100 genes were considered as hit genes. The second step, a hierarchical clustering was performed using the Ward's method.

### Generation of structural models using AlphaFold-Multimer

The Structural models of protein complexes were generated by ColabFold (version 1.4.0) with the following settings: template_mode: none, msa_mode: MMseqs2 (UniRef+Environmental), pair_mode: unpaired + paired, model_type: AlphaFold2-multimer-v2, num_recycles: 3. PDB files and PAE plots of the structural models ranked first among five models were selected as representatives. All representative PDB files were modified and exported by UCSF Chimera (version 1.16).

### Statistics and reproducibility

Statistical analyses of the data were conducted using GraphPad Prism (version 9.3.1) and Microsoft Office Excel (version 2401). $P$ values were determined by the appropriate tests, described in the respective figure legends. Each experiment showing only representative images was repeated independently with similar results at least twice.

### Reporting summary

Further information on research design is available in the Nature Portfolio Reporting Summary linked to this article.

## Data availability

All data used to generate graphs and all pictures showing uncropped gels or blots accompany this manuscript in the source data file. Underlying image data are available from the corresponding authors on reasonable request. The Chronos scores analyzed in this study

(DepMap Public 21Q3 CRISPR_gene_effect) are available on the DepMap website [https://depmap.org/portal/download/all/?releasename=DepMap+Public+21Q3&filename=CRISPR_gene_effect.csv]. The RNA-seq data[50] reanalyzed in this study are available in the NCBI Gene Expression Omnibus (GEO) database under the accession codes GSE60570. Source data are provided with this paper.

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

## Acknowledgements

We thank T. Kawamura, A. Nakayama, Y. Tokunaga, H. Yanagisawa, C. Saito and M. Kikkawa for technical advice; M. Genova for proofreading; The IRCN Imaging Core, The University of Tokyo Institutes for Advanced Studies, D. Kawaguchi and Y. Gotoh for supports of the STED imaging; C. Saito and Tokai Electron Microscopy, Inc. for supports of the TEM imaging; The Broad Institute for providing the DepMap dataset; O. Nureki and all members of Kitagawa laboratory for fruitful discussion. This work was supported by the Japan Society for the Promotion of Science (19H05651 [D.K.], 20K22701 [S.Ha.], 21H02623 [S.Ha.], 22K19305 [M.F.], 21J21492 [Y.T.], 22K19370 [S.Ha.], 23H02627 [T.C.], 23H00394 [T.T.]), Japan Science and Technology Agency (JPMJCR22E1 [D.K.], JPMJPR21EC [S.Ha.]), Takeda Science Foundation [D.K., T.C., S.Y.], Princess Takamatsu Cancer Research Fund [D.K., S.Y.], Uehara Memorial Foundation [T.C., S.Y.], Inamori Foundation [S.Y.], Naito Foundation [T. C.], Kanae Foundation for the Promotion of Medical Science [T.C.], Kato Memorial Bioscience Foundation [S. Hata], Mochida Memorial Foundation for Medical and Pharmaceutical Research [T.C., S.Ha.], Nakajima Foundation [T.C.], and Sumitomo Foundation [T.C.].

## Author contributions

Conceptualization: Y.T., T.C., S.Ha. and D.K.; Methodology: Y.T., T.C., S.T., K.T., T.T., S.Ha. and D.K.; Formal analysis: Y.T.; Investigation: Y.T., S.Ho., S.T. and S.O.; Data curation: Y.T., S.Ho. and S.O.; Visualization: Y.T.; Writing (original draft), Y.T.; Writing (modification): T.C., S.Ha. and D.K.; Supervision: T.C., S.Ha. and D.K. Project administration: Y.T., T.C., S.Ha. and D.K.; Funding acquisition: Y.T., T.C., S.Y., M.F., T.T., S.Ha. and D.K.

## Competing interests

The authors declare no competing interests.
