## [Peer Review File · Nature Communications]

Molecular basis promoting centriole triplet microtubule assemblyREVIEWER COMMENTS

Reviewer #1 (Remarks to the Author):

In the presented manuscript Takeda et al., present novel insights into the long-standing questions about the mechanism of the (in other tubulin-containing structures) uncommon stability of the centriole organelle and centriole-specific microtubule triplet formation.

Using hierarchical clustering of Chronos scores, the authors found HYLS1 in a cluster with CEP295 and CEP44, two genes previously reported to be required for centriole stability.

HYLS1 was previously identified to be required for the formation of fully functional cilia and elongation of the giant centrioles in drosophila spermatocides.

Upon overexpression of HYLS1, the authors observed tubulin- and HYLS1-containing “superstructures” in the cell. The tubulin in these structures, which appeared as cylinders (in some respects similar to centrioles) using STED microscopy had similar properties in stability and resistance assays as centriolar microtubules. Using CLEM, the authors provided evidence that is consistent with the hypothesis that these superstructures contain arrays of unclosed microtubules.

The authors demonstrated that specifically the C-terminus of HYLS1 is required for “superstructure” formation and showed that the D211G mutant, which is found in hydrolethalus patients is required for normal centriole stability and fails to form the tubulin-containing “superstructures” upon overexpression.

Using AlphaFold-Mutimer the authors screened through all alpha- and beta-tubulin genes for potential interaction with HYLS1 and identified one alpha- (TUBA1A) and one beta-tubulin (TUBB5) as potential interaction partner. Moreover, they screened by RNAi through all alpha- and beta-tubulin genes for the requirement for “superstructure” establishment upon HYLS1 overexpression and found only TUBB5 to be required, which they also could Co-IP with HYLS1 from HEK cell lysates. The authors could demonstrate that among all tubulin genes only TUBB5 scores in an STLC centriole stability assay and showed that TUBB5, similar to HYLS1, is required for normal centriole integrity and cilia formation.

Based on their structural predictions, as well of sequence alignment with other beta-tubulin isoforms, they generated a C-terminal truncation and a G437E mutant of TUBB5 (which renders TUBB5 more similar to other beta-tubulin isoforms and is the closest AA in proximity of D211 of HYLS-1 in AlphaFold-Mutimer predictions). While overexpression of the

C-terminal truncation alone was sufficient to form tubulin-containing “superstructures” in the cell, the G437E mutant failed to form these entities, even upon overexpression of HYLS1. Moreover, the G437E mutant failed to rescue centriole stability in an STLC assay. These observations are consistent with the hypothesis put forward by the authors that physical interaction of HYLS1 with the C-terminus of specifically TUBB5 renders this beta-tubulin isoform competent to contribute to tubulin-containing “superstructures” and centriole stability.

The experiments in the submitted manuscript are elegantly designed and appear to be well conducted.

To my mind, the findings are of high significance to the centrosome, cilia and microtubule fields and provide a substantial advance in our to-date knowledge and will inform a substantial amount of future in vivo, in vitro and in silico investigation. They also line-up well with previous studies and suggested mechanisms based on those.

However, to my mind, certain conclusions, especially regarding the specific effects on triplet microtubules in the centriole are not warranted by the data. Those claims should be reworded or supported by additional data (see below).

Overall, I am enthusiastic about the provided manuscript and strongly support publication after addressing the few points below.

Major comments:

- While it is perfectly reasonable to speculate based on the data and AlphaFold prediction presented here that “D211 of HYLS1 physically traps the wobbling C-terminal tail of TUB5, thereby suppressing its inhibitory role ...”, I would not state this as a fact (especially not in the abstract) without further structural evidence. And also, not in several other passages in the main text and discussion.

- I am not sure, how the tube-like “superstructures” observed by STED shown in Figure 1i connect to the line-like structures (organized in half-circles) observed by TEM and shown in Figure 2b and Figure S3b (are these spirals?). Could the authors comment on this in the

main text?

- Given the resolution of the EM images provided, how can the authors be sure to look at incomplete MT, since we don't see protofilaments and (at least on the images in the manuscript) quite heterogenous structures. I think the interlayer distance is a good argument. But, if possible, it would be highly advantageous if higher resolution images could be provided.

- I think that is important: The authors talk at several places specifically about HYLS1s' contribution to stabilize/form TRIPLET microtubules. While this could be the case, to my understanding, these are not established facts. If I understand correctly, the assumption stems from the fact that the "superstructures" observed upon HYLS1 overexpression contain tubulin and electron-dens structures that could be unclosed MTs (like the B- and C-MT in centrioles). And that centriole integrity is impaired in a similar manner as observed in epsilon- or delta-tubulin knockout lines (e.g. Figure 2d). But in the presented examples, half the microtubule wall is missing and not just the B- and C-microtubule. What rules out that HYLS1 does not have another role in centriole formation/stability (e.g. via interaction with CPAP/SAS-4 or anything else)? Also, how do the "narrow centrioles" connect to the role of establishing triplet MTs. According to my reading of the manuscript, I should see in Figure 2d 9 singlet microtubules and not just messed-up centrioles. The authors should clarify this in the text at all relevant positions.

- If the authors did perform any of the following experiments, which could strongly help to support their hypothesis (as far as I understood it) and/or give experimentally mechanistic insights into the proposed role of HYLS1/TUBB5 (supportive or un-supportive for the presented hypothesis) it would be desirable to include those data into the manuscript: 1) Swap out the tail of another beta-tubulin with the TUBB5 tail and looked if it behaves similar to TUBB5 in overexpression experiments; 2) Consequences of overexpression of a beta-tubulin C-terminal truncation mutants (other than TUBB5) on the formation of "superstructures); 3) Assessing the consequence of TUBB5(G437E) on interaction with HYLS1 by CoIP (or similar...)

Minor comments:

- I have some very strange arrays of capital letters instead of some sections in the supplementary figure legends and some supplementary figures are blacked out. Thus, I

could not fully review those sections.

- Line 26: the "...central structure of cilia, flagella and centrosomes." This sounds like an oversimplification as the structural role of the centriole(s) in cilia and centrosome are very different.

- Line 61: The title I feel is too general. I would suggest adding "overexpression of HYL51 leads to assembly of..."

- Figure 1e: in the boxed region of the cluster there appears to be 5 genes. Could the authors spell out their identity?

Reviewer #2 (Remarks to the Author):

A defining feature of cilia and flagella, conserved across eukaryotes, is their being built around a backbone of 9 doublet microtubules, arranged in a radially symmetric manner. Those doublet microtubules emanate from a centriole-derived basal body that itself possesses doublet or triplet microtubules. Unusually, the additional B and C tubules are not composed of 13 protofilaments, but rather are incomplete arcs of 10 protofilaments adhered to the wall of the preceding A and B tubule. How those doublet and triplet microtubules form remains poorly understood, although previous *in vitro* work has implicated modification of the C-terminal tail of tubulin in this process. Here, Takeda and co-workers identify the ciliopathy protein HYL51 as a protein involved in promoting doublet/triplet microtubule assembly *in vivo* through interactions with a specific isoform of beta-tubulin, TUBB5/TUBB. Overall, this study represents a significant conceptual advance, both in our understanding of centriole assembly and the specific functional contribution of HYL51. As such I fully support publication in Nature Communications, although I would ask the authors to address the following points.

Points requiring further experimentation/analysis

1. It is somewhat surprising that the tubulin superstructures induced by overexpression of HYL51 do not recruit any of the centriolar/pericentriolar material proteins examined in Figure S2e. One protein that might perhaps have been expected to co-assemble into those structures is CPAP, whose ortholog in *C. elegans*, SAS-4, recruits HYL51 to centrioles through direct protein-protein interaction (Dammermann, Genes Dev 2009) and which has been

shown to interact with both tubulin and microtubules in numerous studies in vertebrates (eg Hung, Mol Biol Cell 2004; Hsu, Exp Cell Res 2008; Sharma, Dev Cell 2016). Especially given that the immunofluorescence images in this panel were obtained by formaldehyde fixation which frequently masks centrosomal epitopes, what would be essential to show is that the antibodies used here are still capable of recognizing their respective target at centrosomes in the same cells.

2. Based on the data presented I am not fully convinced that TUBB5/TUBB is the sole beta-tubulin isotype involved in doublet/triplet microtubule assembly. The AlphaFold Multimer prediction does point at TUBB (as well as TUBA1A) as potential HYL51 interaction partners. However, the loss of HYL51 overexpression-induced microtubule superstructures only with TUBB depletion may simply reflect the relative abundance of the various tubulin isoforms. If TUBB is the main beta-tubulin isotype in RPE-1 cells, while there is no one alpha-tubulin isotype predominating, this would be the expected result without indicating a function in 2x/3x MT assembly exclusively for TUBB5. My question, then, is what is the abundance of the various alpha/beta-tubulin isotypes in those cells? If TUBB is not the sole beta-tubulin involved in 2x/3x assembly, this would resolve another apparent conundrum, the lack of conservation of this isotype beyond vertebrates (see below).

3. Related to Figure 2, it would be important to compare the geometry of HYL51-induced superstructures to that of centriolar 2x/3x microtubules, similar to what was done in Fig. 2H-J of Schmidt-Cernohorska, Science 2019, to support the authors' contention that these are indeed related structures.

Literature/Discussion points

4. It is not correct to state that centrioles must necessarily possess triplet microtubules for their stability or the ability to assemble functional centrosomes (page 2, 3, 8 and following). In early embryos of *C. elegans* and *Drosophila* centrioles possess singlet or doublet microtubules, respectively, clearly without compromising their stability (see Balestra, Cell Res 2015) or ability to recruit/maintain pericentriolar material. Rather, the primary significance of doublet microtubules is that they are a precondition for templating a doublet axoneme. It is worth noting in this context that later developmental stages of both *C.*

C. elegans and *Drosophila* do assemble basal bodies with doublet/triplet microtubules (Nechipurenko, eLife 2017; Serwas, JCB 2017; Gottardo, JCS 2015). This ought to be clarified in the text.

5. Without wishing to diminish the contribution of Serrano, PNAS 1984, the landmark study reporting the generation of doublet microtubules in vitro by Schmidt-Cernohorska, Science 2019 involved more than the molecular dynamic simulations it is given credit for in the present manuscript and should be cited appropriately.

6. While I do not doubt the involvement of HYLS1 in assembly of centriolar doublet/triplet microtubules, HYLS1 alone clearly cannot explain formation of such microtubule assemblies given that this protein is not visibly conserved beyond metazoans (TUBB5/TUBB may have an even more restricted phylogenetic distribution, being seemingly restricted to vertebrates). HYLS1 being localized exclusively to centrioles/basal bodies, it furthermore cannot explain the existence of axonemal microtubule doublets. There is thus likely a number of proteins involved in modulating the C-terminal tail of tubulins, with HYLS1 only part of the answer. Given that this study should stimulate research to look for these additional proteins, this is a point well worth making.

7. It is worth referencing the work of Woglar, PLOS Biology 2022, which places HYLS-1 just outside the centriolar singlet microtubule in the *C. elegans* germline, the very position a protein involved in doublet/triplet microtubules would be predicted to localize.

8. The authors might want to more fully discuss the work on HYLS-1 in *C. elegans*, which did not just reveal a role for this protein in ciliogenesis, but found it to be fully dispensable for centriole assembly and centrosome function in the early embryo (Dammermann, Genes Dev 2009). Given that centrioles in the early embryo contain singlets whereas basal bodies in ciliated neurons contain doublets (see above), this is perfectly consistent with the authors' proposed role for HYLS1 in centriolar 2x/3x MT formation.

9. It may be interesting to speculate why microtubule superstructures have variable lengths but those ends are sharply defined, ie none of the incomplete microtubules is seemingly

longer/shorter than its neighbors.

Minor points

10. TUBB5 according to HGNC has been renamed TUBB and should be referred to as such.

11. Related to figure S7j, does TUBB localize to cilia as well as centrioles?

12. What the authors mean by interlayer distance (Fig 2c) isn't entirely clear and would be best explained in a schematic.

Reviewer #3 (Remarks to the Author):

This is an excellent study of the ciliopathy-related gene HYLS1 that is required for centriole formation. The authors show that overexpression of the HYLS1 gene results in the formation of tubulin-containing rod like structures. EM analysis revealed these to comprise linked unclosed microtubules of similar width to incomplete ciliary microtubules. Depletion of HYLS1 resulted in shorter centrioles with partially broken or unorganized structures and missing B and C tubules. These cells had abnormal mitotic spindles and could not make primary cilia. A predicted interaction of HYLS1 with tubulin isotype TUBB5 was demonstrated experimentally. Depletion of TUBB5 resulted in a similar phenotype to depletion of HYLS1 and overexpression of a C-terminally truncated, but not full length, TUBB5 induced the formation of open tubulin rods without requiring HYLS1. AlphaFold modelling of the interaction between HYLS1 and TUBB4 predicted the TUBB5 residues that would lie near the D211 residue, frequently mutated in HYLS1 ciliopathy, and mutation of this residue in TUBB5 had a dominant negative effect on assembly of open tubulin rods. These results suggest that HYLS1 acts to suppress the inhibitory effect of the C-terminal tail of beta tubulin upon the initiation of incomplete B and C tubule assembly.

This is a very nice study in which the data support the conclusion being made. It advances our understanding of the assembly of stable centriolar microtubules with the characteristic B-, C- tubule linkages and is highly suited for publication in Nature Comms in its current form.

We would like to thank all three reviewers for the very positive and constructive comments (typed in blue). We have carefully modified our manuscripts with substantial new data and addressed all of their comments as outlined below (typed in black). We are confident that our manuscript is now ready for publication in *Nature Communications*.

Reviewer 1:

The experiments in the submitted manuscript are elegantly designed and appear to be well conducted.

To my mind, the findings are of high significance to the centrosome, cilia and microtubule fields and provide a substantial advance in our to-date knowledge and will inform a substantial amount of future in vivo, in vitro and in silico investigation. They also line-up well with previous studies and suggested mechanisms based on those.

However, to my mind, certain conclusions, especially regarding the specific effects on triplet microtubules in the centriole are not warranted by the data. Those claims should be re-worded or supported by additional data (see below).

Overall, I am enthusiastic about the provided manuscript and strongly support publication after addressing the few points below.

We greatly appreciate the very positive comment that our findings are highly significant and contribute substantially to advancing our knowledge, thereby inspiring future investigations.

Major comments:

- While it is perfectly reasonable to speculate based on the data and AlphaFold prediction presented here that “D211 of HYLS1 physically traps the wobbling C-terminal tail of TUB5, thereby suppressing its inhibitory role ...”, I would not state this as a fact (especially not in the abstract) without further structural evidence. And also, not in several other passages in the main text and discussion.

We are grateful to this reviewer for pointing this out. In response, we have toned down our expressions related to this point throughout the manuscript. In the revised abstract, we now state, “AlphaFold-based structural models and following mutagenesis analyses further *suggest* that the ciliopathy-related residue D211 of HYLS1 physically traps ~” (p.2 line 8). Additionally, the statements listed below have also been toned down.

p.4 line 7 “~, which *physically modulates* the ~” to “~, which *modulates* the ~”

p.4 line 10 “~ analyses *demonstrate* that ~” to “~ analyses *suggest* that ~”

p.16 line 6 “~, which *physically modulates* the ~” to “~, which *modulates* the ~”

p.16 line 16 “~ HYLS1 *physically interacts* with ~” to “~ HYLS1 *interacts* with ~”

p.17 line 23 “~ is *physically modulated* by ~” to “~ is *modulated* by ~”

- I am not sure, how the tube-like “superstructures” observed by STED shown in Figure 1i connect to the line-like structures (organized in half-circles) observed by TEM and shown in Figure 2b and Figure S3b (are these spirals?). Could the authors comment on this in the main text?

We apologize for any confusion caused by the presentations regarding the superstructures in the initial version. As requested by this reviewer, we have revised the manuscript to provide more details (p.7 line 17 & line 20) and included a new schematic related to the TEM images (Supplementary Figure 4b) for clearer image correlation.

- Given the resolution of the EM images provided, how can the authors be sure to look at incomplete MT, since we don't see protofilaments and (at least on the images in the manuscript) quite heterogenous structures. I think the interlayer distance is a good argument. But, if possible, it would be highly advantageous if higher resolution images could be provided.

We thank this reviewer for the helpful suggestion. According to this comment, we have conducted various experiments to achieve higher resolution images of the top-viewed incomplete microtubules within the superstructures. In this process, following advice from electron microscopy experts, we have tried to optimize our protocol by 1) improving fixatives (e.g., adding tannic acid), 2) using TEM models with higher accelerating voltage (90 kV to 100-120 kV), 3) out-sourcing of professional analyses (Tokai Electron Microscopy, Inc.). Despite our efforts, we were unable to capture images with sufficient resolution to discern protofilaments. Nevertheless, we believe our conclusion is robust based on the morphology and dimensions of the top-viewed images (Figure 2a and Supplementary Figure 4c) and the analysis of the interlayer distances using the side-viewed images (Figure 2b, c).

- I think that is important: The authors talk at several places specifically about HYLS1s' contribution to stabilize/form TRIPLET microtubules. While this could be the case, to my understanding, these are not established facts. If I understand correctly, the assumption stems from the fact that the “superstructures” observed upon HYLS1 overexpression contain tubulin and electron-dens structures that could be unclosed MTs (like the B- and C-MT in centrioles). And that centriole integrity is impaired in a similar manner as

observed in epsilon- or delta-tubulin knockout lines (e.g. Figure 2d). But in the presented examples, half the microtubule wall is missing and not just the B- and C-microtubule. What rules out that HYLS1 does not have another role in centriole formation/stability (e.g. via interaction with CPAP/SAS-4 or anything else)? Also, how do the “narrow centrioles” connect to the role of establishing triplet MTs. According to my reading of the manuscript, I should see in Figure 2d 9 singlet microtubules and not just messed-up centrioles. The authors should clarify this in the text at all relevant positions.

We appreciate this reviewer for the constructive comment. Indeed, as this reviewer pointed out, the centriole structures in HYLS1 knockdown cells (Figure 2d) appear, at first glance, to be inconsistent with the presumed function of HYLS1 in promoting the assembly of incomplete microtubules. Nonetheless, our conclusion remains unchanged for the following reasons. The simultaneous occurrence of incomplete tubule loss and centriole wall collapse (similar to Figure 2d; Partial defect) shown in **Figure 8c** of Vasquez-Limeta *et al.*, *J. Cell. Biol.* 2022 suggests that intact triplet microtubules are crucial for maintaining centriole wall integrity. We also observed centrioles composed of singlet-like microtubules without accompanying wall collapse (Figure 2d; Severe defect). Therefore, the observed centriole wall collapse might be a secondary effect of the incomplete tubule loss. Moreover, the revised manuscript now includes new insights regarding the difference in need for HYLS1 in singlet-based centrioles and doublet-based cilia in *C. elegans*, which substantially reinforces our conclusion (p. 8 line 12-15). Although it is conceivable that HYLS1 may also contribute to centriole formation or stability through other mechanisms, such considerations are beyond the scope of this paper.

- If the authors did perform any of the following experiments, which could strongly help to support their hypothesis (as far as I understood it) and/or give experimentally mechanistic insights into the proposed role of HYLS1/TUBB5 (supportive or un-supportive for the presented hypothesis) it would be desirable to include those data into the manuscript:

We are grateful to this reviewer for the valuable suggestions. In response to the recommendation to conduct additional experiments to reinforce our hypothesis, we have carried out all the suggested experiments. The results and interpretations corresponding to each comment are described below.

1) Swap out the tail of another beta-tubulin with the TUBB5 tail and looked if it behaves similar to TUBB5 in overexpression experiments;

As suggested, we have newly created cell lines expressing

- 1) TUBB4B (a major β -tubulin isotype along with TUBB)
 - 2) TUBB swapped its body into that of TUBB4B (TUBB4B-TUBB CTT)
 - 3) TUBB swapped its CTT into that of TUBB4B (TUBB-TUBB4B CTT)
- (Supplementary Fig. 13a-c).

Then, rescue experiments similar to those in Figure 6f, g have been performed with these cell lines, revealing that none of them can reverse the reduction in cells with superstructures upon siTUBB treatment (Supplementary Figure 14a, b). These results indicate that both the body and the CTT of TUBB are crucial for the HYLS1-dependent assembly of incomplete microtubules.

2) Consequences of overexpression of a beta-tubulin C-terminal truncation mutants (other than TUBB5) on the formation of “superstructures”;

Following the suggestion, we have overexpressed a CTT truncated version of TUBB4B, which led to the formation of similar superstructures (Supplementary Figure 12c, d). However, given that β -tubulin isotypes other than TUBB did not exhibit strong interactions with HYLS1 in the AlphaFold-Multimer predictions (Figure 4c) and were dispensable for the HYLS1-dependent superstructure formation (Figure 4f), it seems likely that the TUBB body is specifically required for the HYLS1-dependent assembly of superstructures.

3) Assessing the consequence of TUBB5(G437E) on interaction with HYLS1 by CoIP (or similar...)

As recommended, we have performed Co-IP experiments with TUBB G437E (a) and HYLS1 D211G (b) mutants (attached below), revealing that these mutants can also interact with their respective partners. These results indicate that the regulation of TUBB CTT by HYLS1 itself is dispensable for the direct interaction between the two proteins.

Minor comments:

- I have some very strange arrays of capital letters instead of some sections in the supplementary figure legends and some supplementary figures are blacked out. Thus, I could not fully review those sections.

We are very sorry for any inconvenience during the review process. The revised version has been prepared with care to avoid such issues.

- Line 26: the "...central structure of cilia, flagella and centrosomes." This sounds like an oversimplification as the structural role of the centriole(s) in cilia and centrosome are very different.

Thank you very much for your insightful comment. In response, we have carefully reworded the sentence in the introduction to avoid oversimplification as follows.

p.3 line 2 "~ is *central structure* of ~" to "~ is *essential for the formation* of ~"

- Line 61: The title I feel is too general. I would suggest adding "overexpression of HYLS1 leads to assembly of..."

We would like to thank this suggestion. As suggested, we have modified the titles of the relevant section and figures to begin with "*Overexpression of HYLS1 leads to ~*" (please see p.5 line 2 and the legends of Figure 1 and Supplementary Figure 2-4).

- Figure 1e: in the boxed region of the cluster there appears to be 5 genes. Could the authors spell out their identity?

As requested by this reviewer, we have spelled out the full gene names in both Figure 1 and the manuscript (p.5 line 20-21).

Reviewer 2:

A defining feature of cilia and flagella, conserved across eukaryotes, is their being built around a backbone of 9 doublet microtubules, arranged in a radially symmetric manner. Those doublet microtubules emanate from a centriole-derived basal body that itself possesses doublet or triplet microtubules. Unusually, the additional B and C tubules are not composed of 13 protofilaments, but rather are incomplete arcs of 10 protofilaments adhered to the wall of the preceding A and B tubule. How those doublet and triplet microtubules form remains poorly understood, although previous *in vitro* work has implicated modification of the C-terminal tail of tubulin in this process. Here, Takeda and co-workers identify the ciliopathy protein HYLS1 as a protein involved in promoting doublet/triplet microtubule assembly *in vivo* through interactions with a specific isoform of beta-tubulin, TUBB5/TUBB. Overall, this study represents a significant conceptual

advance, both in our understanding of centriole assembly and the specific functional contribution of HYLS1. As such I fully support publication in Nature Communications, although I would ask the authors to address the following points.

We thank this reviewer for the very supportive comment that this study represents a significant conceptual advance in several aspects.

Points requiring further experimentation/analysis

1. It is somewhat surprising that the tubulin superstructures induced by overexpression of HYLS1 do not recruit any of the centriolar/pericentriolar material proteins examined in Figure S2e. One protein that might perhaps have been expected to co-assemble into those structures is CPAP, whose ortholog in *C. elegans*, SAS-4, recruits HYLS1 to centrioles through direct protein-protein interaction (Dammermann, *Genes Dev* 2009) and which has been shown to interact with both tubulin and microtubules in numerous studies in vertebrates (e.g. Hung, *Mol Biol Cell* 2004; Hsu, *Exp Cell Res* 2008; Sharma, *Dev Cell* 2016). Especially given that the immunofluorescence images in this panel were obtained by formaldehyde fixation which frequently masks centrosomal epitopes, what would be essential to show is that the antibodies used here are still capable of recognizing their respective target at centrosomes in the same cells.

We thank this reviewer for pointing this out. To address this concern, we have performed an additional experiment (Supplementary Figure 3c). In this experiment, we fixed HYLS1 overexpressed cells with ice-cold methanol and subsequently stained them with antibodies against α -tubulin (Green; for superstructures), Centrin (Gray; for centrioles), and centrosomal proteins (Magenta; same as in Supplementary Figure 3b). This analysis revealed that these centrosomal proteins, including CPAP, were not present on the tubulin-based superstructures, although their epitopes were clearly unmasked at the centrioles.

2. Based on the data presented I am not fully convinced that TUBB5/TUBB is the sole beta-tubulin isotype involved in doublet/triplet microtubule assembly. The AlphaFold Multimer prediction does point at TUBB (as well as TUBA1A) as potential HYLS1 interaction partners. However, the loss of HYLS1 overexpression-induced microtubule superstructures only with TUBB depletion may simply reflect the relative abundance of the various tubulin isoforms. If TUBB is the main beta-tubulin isotype in RPE-1 cells, while there is no one alpha-tubulin isotype predominating, this would be the expected result without indicating a function in 2x/3x MT assembly exclusively for TUBB5. My question, then, is what is the abundance of the various alpha/beta-tubulin isotypes in those cells? If TUBB is not the sole beta-tubulin involved in 2x/3x assembly, this would resolve

another apparent conundrum, the lack of conservation of this isotype beyond vertebrates (see below).

Thank you very much for pointing this out. In response, we have analyzed the expression levels of the tubulin isotypes using a previously published RNA-seq dataset of RPE-1 cells (Supplementary Figure 8a). In this analysis, the expression of TUBB4B was comparable to that of TUBB, and TUBA1B was predominantly expressed among the α -tubulin isotypes. Additionally, we have assessed gene knockdown efficiency of the major tubulin isotypes in RPE-1 cells (Supplementary Figure 8b). Furthermore, our new results indicate that TUBB4B cannot compensate for the TUBB knockdown phenotype (Supplementary Figure 14a, b). Taken together, the selective suppression of superstructure assembly upon TUBB knockdown is *not* due to the relative abundance of the tubulin isotypes.

3. Related to Figure 2, it would be important to compare the geometry of HYLS1-induced superstructures to that of centriolar 2x/3x microtubules, similar to what was done in Fig. 2H-J of Schmidt-Cernohorska, *Science* 2019, to support the authors' contention that these are indeed related structures.

We appreciate this reviewer for the insightful comment. As suggested, we have attempted to analyze the geometry of the superstructures, but we faced challenges due to the heterogeneity of the incomplete tubules. This heterogeneity is likely because each incomplete tubule nucleates from another incomplete tubule. By contrast, Schmidt-Cernohorska *et al*, *Science*, 2019, analyzed incomplete tubules nucleating from pre-existing complete microtubules, which enabled them to acquire more uniform data.

Literature/Discussion points

4. It is not correct to state that centrioles must necessarily possess triplet microtubules for their stability or the ability to assemble functional centrosomes (page 2, 3, 8 and following). In early embryos of *C. elegans* and *Drosophila* centrioles possess singlet or doublet microtubules, respectively, clearly without compromising their stability (see Balestra, *Cell Res* 2015) or ability to recruit/maintain pericentriolar material. Rather, the primary significance of doublet microtubules is that they are a precondition for templating a doublet axoneme. It is worth noting in this context that later developmental stages of both *C. elegans* and *Drosophila* do assemble basal bodies with doublet/triplet microtubules (Nechipurenko, *eLife* 2017; Serwas, *JCB* 2017; Gottardo, *JCS* 2015). This ought to be clarified in the text.

Thank you very much for pointing this out. We acknowledge that our initial statement

that triplet microtubules are universally essential for the stability and function of centrioles in all organisms, including *C. elegans* and *Drosophila*, is incorrect. To correct this, we have reworded the relevant sections as follows.

p. 3 line 9 “~ microtubules *enable* the ~” to “~ microtubules *are crucial for* the ~”

p. 3 line 12 “Centrioles lacking the ~” to “*In human cells*, centrioles lacking the ~”

p. 8 line 17 “~ in previous studies” to “~ in previous studies *using human cells*”

p. 9 line 3 “~ is *crucial* for ~” to “~ is *important* for ~”

5. Without wishing to diminish the contribution of Serrano, PNAS 1984, the landmark study reporting the generation of doublet microtubules in vitro by Schmidt-Cernohorska, Science 2019 involved more than the molecular dynamic simulations it is given credit for in the present manuscript and should be cited appropriately.

We would like to appreciate this reviewer for pointing this out. In the revised version, we have carefully rephrased the relevant paragraph to appropriately reference the cited studies (please see p. 3 line 18 - p.4 line 5).

6. While I do not doubt the involvement of HYLS1 in assembly of centriolar doublet/triplet microtubules, HYLS1 alone clearly cannot explain formation of such microtubule assemblies given that this protein is not visibly conserved beyond metazoans (TUBB5/TUBB may have an even more restricted phylogenetic distribution, being seemingly restricted to vertebrates). HYLS1 being localized exclusively to centrioles/basal bodies, it furthermore cannot explain the existence of axonemal microtubule doublets. There is thus likely a number of proteins involved in modulating the C-terminal tail of tubulins, with HYLS1 only part of the answer. Given that this study should stimulate research to look for these additional proteins, this is a point well worth making.

Thank you very much for highlighting this important issue. Our interpretation regarding the two topics is as follows. 1) Evolutionary conservation; While it is conceivable that an entirely distinct system may exist in these species for promoting triplet microtubule assembly, it is equally possible that counterparts to HYLS1 and TUBB have not yet been identified in these organisms. 2) Ciliary localization; Although HYLS1 does not localize to cilia, the observed defects in primary cilia elongation in HYLS1 or TUBB knockdown cells (Figure 2m and Figure 5f) indicate a potential role of HYLS1 in the development of ciliary doublet microtubule structures. It is possible that HYLS1 facilitates the assembly of B-tubules within the basal bodies (centrioles), which in turn may serve as templates for autonomous elongation of ciliary B-tubules. Taken together, because of the various

possibilities and the lack of clear evidence, these issues are beyond the scope of this paper. We would like to address them in future research.

7. It is worth referencing the work of Woglar, PLOS Biology 2022, which places HYLS-1 just outside the centriolar singlet microtubule in the *C. elegans* germline, the very position a protein involved in doublet/triplet microtubules would be predicted to localize. We thank this reviewer for recommending the citation of Woglar *et al.*, *PLoS Biol.* 2022 to enhance our discussion. As advised, we now include this study in the discussion section (please see p.17 line 8-9).

8. The authors might want to more fully discuss the work on HYLS-1 in *C. elegans*, which did not just reveal a role for this protein in ciliogenesis, but found it to be fully dispensable for centriole assembly and centrosome function in the early embryo (Dammermann, *Genes Dev* 2009). Given that centrioles in the early embryo contain singlets whereas basal bodies in ciliated neurons contain doublets (see above), this is perfectly consistent with the authors' proposed role for HYLS1 in centriolar 2x/3x MT formation.

We appreciate this reviewer for the valuable suggestion. In accordance with this advice, we have referred to the study in the results section to emphasize the role of HYLS1 in the formation of incomplete tubules, B- and C-tubules (please see p. 8 line 12-15).

9. It may be interesting to speculate why microtubule superstructures have variable lengths but those ends are sharply defined, ie none of the incomplete microtubules is seemingly longer/shorter than its neighbors.

We appreciate this reviewer for highlighting this crucial aspect regarding the superstructures. We now hypothesize that the lateral formation of incomplete tubules occurs at a much faster rate than the elongation of superstructures, which potentially explains the observed phenomenon. Verifying this hypothesis, however, would require time-lapse, high-resolution monitoring of the assembly process of superstructures, a task beyond the scope of this study. We plan to investigate this in future research.

Minor points

10. TUBB5 according to HGNC has been renamed TUBB and should be referred to as such.

Thank you very much for pointing this out. In the revised version, we have updated all “TUBB5” to “TUBB” in both the manuscript and the figures.

11. Related to figure S7j, does TUBB localize to cilia as well as centrioles?

We thank this reviewer for raising an important question about TUBB localization. As shown in Supplementary Figure 15f (upon siTUBB treatment) and additional images attached below (without siRNA treatment), overexpressed TUBB also localizes to primary cilia in RPE-1 cells.

12. What the authors mean by interlayer distance (Fig 2c) isn't entirely clear and would be best explained in a schematic.

We apologize for any previous lack of clarity regarding the “interlayer distance” quantified in Figure 2c. In the revised version, we have added a new schematic for a clearer definition of “interlayer distance” (Supplementary Figure 4b).

Reviewer 3:

This is a very nice study in which the data support the conclusion being made. It advances our understanding of the assembly of stable centriolar microtubules with the characteristic B-, C- tubule linkages and is highly suited for publication in *Nature Comms* in its current form.

We deeply thank this reviewer for the very supportive comment that our manuscript was already suitable for publication in *Nature Communications*.

REVIEWERS' COMMENTS

Reviewer #1 (Remarks to the Author):

The authors have tried to (in my assessment in good faith) or did address all my concerns. I remain to think that this is a beautiful study and I hope to see it soon published in the targeted journal.

Reviewer #2 (Remarks to the Author):

In preparing this revision Takeda and co-workers overall have done an excellent job addressing the points made by myself and the other reviewers, including by significant additional experimentation. I therefore fully support publication in Nature Communications, although I would still encourage the authors to incorporate the following text changes:

1. Related to original points 2/6, if a C-terminal tail-truncated version of another β -tubulin isotype, TUBB4B, also induces assembly of superstructures (Fig S12, p13), even if TUBB4B is dispensable for HYLS1-induced superstructure formation would this not point at other tubulins besides TUBB and other proteins besides in HYLS1 in centriolar/ciliary doublet formation? Especially given the limited restricted phylogenetic distribution of TUBB/HYLS1 compared to the pan-eukaryote conservation of centriolar doublet/triplet microtubules, this seems highly likely. Conceding that point and allowing for the possible existence of other players would not in any way diminish from the authors' findings while stimulating further work. I strongly encourage the authors to include such an evolutionary perspective in their discussion.

2. Related to original points 4/8, centrioles with singlet microtubules are only found in in *C. elegans* early embryos. The centrioles with doublet microtubules which will give rise to cilia in ciliated sensory neurons also form in embryos, albeit a few hours later. To avoid confusion the sentence on p8 should be modified as follows: "In contrast, centrioles in *C. elegans* early embryos, which are composed of singlet microtubules, do not require HYLS1 for their assembly."

We would like to thank the two reviewers for the very positive and constructive comments (typed in blue). We have carefully modified our manuscripts and addressed all of their comments as outlined below (typed in black). We have also formatted the main manuscript, figures, and reporting summary according to the editorial requests.

Reviewer 1:

The authors have tried to (in my assessment in good faith) or did address all my concerns. I remain to think that this is a beautiful study and I hope to see it soon published in the targeted journal.

We deeply thank this reviewer for the very supportive comment that our manuscript is already suitable for publication in *Nature Communications*.

Reviewer 2:

In preparing this revision Takeda and co-workers overall have done an excellent job addressing the points made by myself and the other reviewers, including by significant additional experimentation. I therefore fully support publication in Nature Communications, although I would still encourage the authors to incorporate the following text changes:

We would like to appreciate this reviewer for supporting publication in *Nature Communications*.

1. Related to original points 2/6, if a C-terminal tail-truncated version of another β -tubulin isotype, TUBB4B, also induces assembly of superstructures (Fig S12, p13), even if TUBB4B is dispensable for HYLS1-induced superstructure formation would this not point at other tubulins besides TUBB and other proteins besides in HYLS1 in centriolar/ciliary doublet formation? Especially given the limited restricted phylogenetic distribution of TUBB/HYLS1 compared to the pan-eukaryote conservation of centriolar doublet/triplet microtubules, this seems highly likely. Conceding that point and allowing for the possible existence of other players would not in any way diminish from the authors' findings while stimulating further work. I strongly encourage the authors to include such an evolutionary perspective in their discussion.

Thank you very much for the valuable suggestion. In the revised version, we have added a new paragraph including an evolutionary perspective and an argument about the possible existence of additional systems to the discussion section (please see p. 17 line

12-22).

2. Related to original points 4/8, centrioles with singlet microtubules are only found in in *C. elegans* early embryos. The centrioles with doublet microtubules which will give rise to cilia in ciliated sensory neurons also form in embryos, albeit a few hours later. To avoid confusion the sentence on p8 should be modified as follows: "In contrast, centrioles in *C. elegans* early embryos, which are composed of singlet microtubules, do not require *HYLS1* for their assembly."

We are grateful to this reviewer for pointing this out. We acknowledge that our statement was not appropriate. According to this comment, we have rephrased the sentence as suggested (please see p. 8 line 13-15).